# Latent-Guided Cooperative Energy-Based Models

**Cong Geng** [1]  **Xue Han** [1]  **Ye Yuan** [1]  **Qiang Hu** [2]  **Xin Huang** [1]  **Ruiqiao Bai** [1]  **Junlan Feng** [1]

## Abstract

Energy-based models (EBMs) provide a flexible framework for generative models with strong distribution modeling capabilities. Nevertheless, their broader adoption has been limited by the difficulty of stable and efficient training. In this paper, we propose a unified and efficient latent-guided cooperative EBM that leverages informative target latent variables to guide the joint energy in capturing both data distribution and semantic structure, along with a cooperative generator designed for effective MCMC initialization. Our joint space optimization only requires MCMC sampling in the data space, and allows the energy to learn semantic data–latent relationships directly from real data. Experiments show our method improves generation quality and training stability with fewer resources, and performs effectively across multiple downstream tasks.

## 1. Introduction

Generative models have achieved unprecedented rapid development in recent years. Energy-based models (EBMs) (Le-Cun et al., 2006; Salakhutdinov et al., 2007; Du & Mordatch, 2019), as a class of generative models, occupy a unique position among various generative frameworks due to their huge potential in modeling complex data distributions. With a flexible energy function to directly characterize the underlying probability distribution, EBM can be useful in various tasks such as image and video synthesis (Xie et al., 2019; Zhao et al., 2020), image restoration (Xie et al., 2021a; Gao et al., 2021), compositional generation (Du et al., 2020; 2023), and out-of-distribution (OOD) detection (Yoon et al., 2021; 2023). However, it is notorious for hard training and long-run MCMC sampling (Nijkamp et al., 2020; Grath-

[1]China Mobile Jiutian Artificial Intelligence Technology (Beijing) Co., Ltd [2]Cooperative Medianet Innovation Center, Shanghai Jiao Tong University, Shanghai, China. Correspondence to: Xue Han <hanxueit@chinamobile.com>, Junlan Feng <fengjunlanit@chinamobile.com>.

*Proceedings of the 43rd International Conference on Machine Learning*, Seoul, South Korea. PMLR 306, 2026. Copyright 2026 by the author(s).

wohl et al., 2021), leaving a noticeable gap with dominant generative models such as GANs (Karras et al., 2020) and Diffusion Models (Ho et al., 2020; Song et al., 2021).

Adversarial EBMs (Geng et al., 2021; 2024) and cooperative learning (Xie et al., 2020) incorporate a generator to speed up sampling and improve generation quality. However, adversarial EBMs are prone to suffering from mode collapse because of their minimax training strategy. Cooperative learning leads to biased generator learning, thereby limiting the potential for learning a robust EBM. Divergence Triangle methods (Han et al., 2019; 2020) extend this co-training scheme to latent-variable models. However, by enforcing exact alignment between the latent representation and the generator's prior, they restrict both generation quality and latent space flexibility, ultimately weakening the energy function. CLEL (Lee et al., 2023) designs a new class of latent-variable EBMs that model the joint distribution using a contrastive latent encoder. This architecture enables the energy function to benefit from the semantically latent representations, moving beyond the conventional Gaussian posterior. But its training paradigm is cumbersome, and sampling remains slow.

To mitigate training challenges in EBMs, we propose a cooperative training framework that combines a latent-guided EBM (LGEBM) with auxiliary generator initialization. For each training step, the energy function and generator are updated alternatively. When training the energy function, target latent variables are derived through a pretrained self-supervised latent encoder. With our defined joint energy formulation, we theoretically show that MCMC sampling is only required in the data space, avoiding the expensive cost in joint space. Moreover, this design also enables learning semantic data-latent relationships directly from real data, in a simpler way than previous contrastive learning (Han et al., 2020). Beyond only distribution modeling, our informative latent variables help energy function capture the semantic geometry of the data manifold, as shown in Fig.1. The introduced generator learns to initialize the long-run MCMC dynamics through a single-step transformation, enabling efficient short-run sampling and even one-step generation. An augmentation technique is applied to the generated samples to improve the energy landscape. Our method enhances EBM performance through joint energy optimization with latent guidance under a dedicated cooperative training

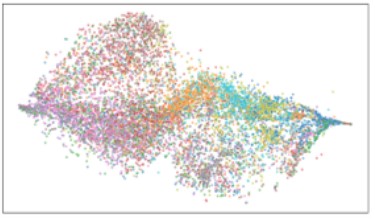

(a) $f_\phi(x)$ of cooperative EBM

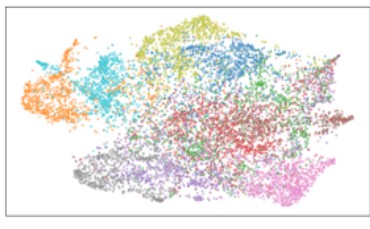

(b) $f_\phi(x)$ of LGCEBM

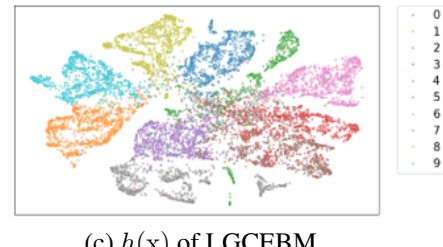

(c) $h(x)$ of LGCEBM

*Figure 1.* t-SNE visualization trained on CIFAR-10: cooperative EBM with $F(f_\phi(x))$ as energy function *vs.* our LGCEBM.

framework. Furthermore, it decouples the generative prior from the semantic latent representation, thus avoiding the potential pitfalls of "posterior collapse" (Geng et al., 2023). Our main contributions are summarized as follows:

1. We propose a unified and efficient latent-guided EBM that highlights the importance of joint space optimization. We utilize pretrained self-supervised representations as target latent variables to guide the energy. Our training paradigm requires MCMC sampling only in data space, enabling the joint energy to learn semantic data–latent relationships directly from real samples.

2. We develop an efficient cooperative training framework between LGEBM and an auxiliary generator, incorporating negative-sample augmentation and adaptive generator training designs.

3. Our method achieves superior generation quality with reduced computational cost, while exhibiting versatile applicability across multiple downstream tasks.

## 2. Preliminary

Latent-variable EBMs generalize standard EBMs by incorporating a latent variable to model a joint distribution $p_\theta(x, z)$:

$$p_\theta(x, z) = \frac{\exp(-E_\theta(x, z))}{Z_\theta}, Z_\theta = \int \exp(-E_\theta(x, z)) \, dxdz,$$
(1)

where $Z_\theta$ is the intractable normalizing constant called the partition function. Training latent-variable EBMs primarily relies on minimizing the negative log-likelihood such that:

$$\begin{aligned} L(\theta) &:= -\mathbb{E}_{(x,z)\sim p_{\text{data}}(x,z)} [\log p_\theta(x, z)] \\ &= \mathbb{E}_{(x,z)\sim p_{\text{data}}(x,z)} [E_\theta(x, z)] + \log Z_\theta. \end{aligned}$$
(2)

Similar to standard EBMs, the gradient of the training objective can be written as:

$$\begin{aligned} \frac{\partial L}{\partial \theta} &= \mathbb{E}_{(x,z)\sim p_{\text{data}}(x,z)} \left[ \frac{\partial}{\partial \theta} E_\theta(x, z) \right] \\ &- \mathbb{E}_{(x,z)\sim p_\theta(x,z)} \left[ \frac{\partial}{\partial \theta} E_\theta(x, z) \right]. \end{aligned}$$
(3)

It requires MCMC sampling from a joint distribution $p_\theta(x, z)$, which can be challenging in complex high-dimensional space (Xu et al., 2018). See App.B for additional details about EBMs.

## 3. Method

Conventional EBMs train the energy function solely in the data space, which poses challenges in high-dimensional settings due to data sparsity and limited distributional information. To address this, we propose a latent-variable EBM with structured latent guidance and generator-assisted MCMC initialization, called the **L**atent-**G**uided **C**ooperative **EBM** (LGCEBM). The energy function and generator are trained alternatively within each training step. Our formulation needs to solve three fundamental problems: defining a target joint distribution $p_{\text{data}}(x, z)$ given only observed samples $x$; constructing a joint energy distribution $p_\theta(x, z)$ that captures data-latent coupling; and balancing cooperative training between the energy function and generator.

### 3.1. Joint Energy Training

#### 3.1.1. TARGET JOINT DISTRIBUTION

The target joint distribution can be defined in multiple ways, since we only have access to observed data. What needs to be considered is which choice of the conditional latent distribution $p_{\text{data}}(z|x)$ can leverage useful information to facilitate EBM training and be easy to sampling. Accordingly, we define $p_{\text{data}}(z|x)$ by mapping data samples to normalized latent variables through a latent encoder after random transformation, i.e., sampling $p_{\text{data}}(z|x)$ through $h(v(x))/\|h(v(x))\|_2, v \sim \mathcal{V}$, where $\mathcal{V}$ is a random operation distribution. The latent encoder $h$ is pretrained using self-supervised representation learning (SSRL) method as a separate stage before EBM training. We design it this way because SSRL is known to extract meaningful semantic features from the data manifold (Chen et al., 2020; Grill et al., 2020; Caron et al., 2021; Yu et al., 2025), which is expected to provide effective latent guidance for energy training. We consider two options for $\mathcal{V}$:

(i) The standard random augmentations commonly used in SSRL, including image cropping, rotation, color jittering,

Sobel filtering, etc. We use the same augmentations as in our pretrained SSRL.

(ii) Adding minor uniform noise:

$$v(\mathrm{x}) = \frac{255}{256}\mathrm{x} + \mathrm{z}, \quad \mathrm{z} \sim U(0, \frac{1}{256}) \qquad (4)$$

In this way, latent variable sampling approximates to a deterministic function, eliminating the reliance of $p_{\mathrm{data}}(\mathrm{z}|\mathrm{x})$ on hand-crafted augmentations.

Both choices yield similar performance, with the selection depending on the SSRL method. Overall, the target joint distribution is defined via $p_{\mathrm{data}}(\mathrm{x}, \mathrm{z}) = p_{\mathrm{data}}(\mathrm{x})p_{\mathrm{data}}(\mathrm{z}|\mathrm{x})$, where $p_{\mathrm{data}}(\mathrm{x})$ is real data distribution.

### 3.1.2. JOINT ENERGY FUNCTION

Considering modeling a joint energy distribution, we first simplify the traditional training objective in Eq.3 using a marginal energy:

**Proposition 3.1.** *For an energy function defined in Eq.1, let $E_\theta(\mathrm{x}) = -\log \int \exp\left(-E_\theta(\mathrm{x}, \mathrm{z})\right) d\mathrm{z}$, then $E_\theta(\mathrm{x})$ is an available energy function of marginal $p_\theta(\mathrm{x})$ and Eq.3 can be reformulated to:*

$$\frac{\partial L}{\partial \theta} = \mathbb{E}_{(\mathrm{x},\mathrm{z}) \sim p_{data}(\mathrm{x},\mathrm{z})} \left[ \frac{\partial}{\partial \theta} E_\theta(\mathrm{x}, \mathrm{z}) \right] - \mathbb{E}_{\mathrm{x} \sim p_\theta(\mathrm{x})} \left[ \frac{\partial}{\partial \theta} E_\theta(\mathrm{x}) \right],$$
$$(5)$$

See App.C for the proof. Proposition 3.1 yields a simplified training objective that requires MCMC sampling only in data space, avoiding the high cost of joint $(\mathrm{x}, \mathrm{z})$ space. This design also enables direct learning of semantic data–latent relationships from real $p_{\mathrm{data}}(\mathrm{x}, \mathrm{z})$, without relying on contrastive exclusion of fake pairs commonly used in representation learning (Ye et al., 2019) and EBMs (Han et al., 2020). Next, we investigate several forms of $E_\theta(\mathrm{x}, \mathrm{z})$.

**Conjugate Exponential Families** If we express $E_\theta(\mathrm{x}, \mathrm{z})$ in the exponential family form (Wu et al., 2021),

$$E_\theta(\mathrm{x}, \mathrm{z}) = -\langle \lambda + f_\phi(\mathrm{x}), \eta(\mathrm{z}) \rangle + B(\lambda), \qquad (6)$$

where $B(\cdot)$ is the log-partition function and $f_\phi(\mathrm{x})$ is a neural network parameterized by $\phi$. $\eta(\mathrm{z})$ maps latent variables to sufficient statistics. Then $E_\theta(\mathrm{x}) = -\log \int \exp\left(-E_\theta(\mathrm{x}, \mathrm{z})\right) d\mathrm{z}$ can be expressed as:

$$E_\theta(\mathrm{x}) = B(\lambda) - B\left(\lambda + f_\phi(\mathrm{x})\right) \qquad (7)$$

See App.D for details. If we set $\eta(z_k) = \left(z_k, z_k^2\right)$ for each dimension of z, with $\lambda$ and $B(\lambda)$ given by:

$$\lambda = (\lambda_1, \lambda_2) = (0, -0.5), \qquad (8)$$

$$B(\lambda) = -\frac{\lambda_1^2}{4\lambda_2} - \frac{1}{2}\log\left(-2\lambda_2\right) \qquad (9)$$

$p_\theta(\mathrm{z}|\mathrm{x})$ becomes a Gaussian distribution. This joint energy determines both the marginal energy and the latent posterior in a unified form. We can also decompose the joint energy density into an implicit marginal distribution and an explicit latent posterior:

$$E_\theta(\mathrm{x}, \mathrm{z}) = F\left(f_\phi(\mathrm{x})\right) - \log p_{\phi,\psi}(\mathrm{z}|\mathrm{x}), \qquad (10)$$

where $F$ maps $f_\phi(\mathrm{x})$ to a scalar, which can be a nonparametric function or a neural network. $p_{\phi,\psi}(\mathrm{z}|\mathrm{x})$ is a probability density parameterized by $(\phi, \psi)$, and $\theta = (\phi, \psi)$. In this form, $E_\theta(\mathrm{x}) = F\left(f_\phi(\mathrm{x})\right)$. In the following, we consider two choices of $p_{\phi,\psi}(\mathrm{z}|\mathrm{x})$.

**Gaussian Posterior** A natural choice of $p_{\phi,\psi}(\mathrm{z}|\mathrm{x})$ is Gaussian distribution. In this case, we let $f_\phi$ be a flexible neural network, and

$$p_{\phi,\psi}(\mathrm{z}|\mathrm{x}) \sim \mathcal{N}\left(g_{\psi_m}(f_\phi(\mathrm{x})), g_{\psi_\sigma}(f_\phi(\mathrm{x}))\right) \qquad (11)$$

where $\psi = \psi_m \cup \psi_\sigma$. This definition is similar to conventional latent-variable EBMs (Kan et al., 2022; Cui & Han, 2023), except that the marginal energy $E_\theta(\mathrm{x})$ and the posterior $p_{\phi,\psi}(\mathrm{z}|\mathrm{x})$ share the backbone $f_\phi$, coupling them architecturally.

**Cosine-Similarity Posterior** Beyond Gaussian posterior, we also adopt a cosine-similarity form to define the posterior on a unit sphere:

$$p_{\phi,\psi}(\mathrm{z}|\mathrm{x}) = \frac{\exp\left(\gamma \operatorname{sim}\left(g_\psi\left(f_\phi(\mathrm{x})\right), \mathrm{z}\right)\right)}{Z_\gamma}, \quad \mathrm{z} \sim \mathbb{S}^{d_\mathrm{z}-1} \qquad (12)$$

where $\operatorname{sim}(\mathrm{u}, \mathrm{v}) = \mathrm{u}^\top \mathrm{v}/\|\mathrm{u}\|_2\|\mathrm{v}\|_2$ is the cosine similarity. CLEL (Lee et al., 2023) uses a similar definition with normalized $f_\phi(\mathrm{x})$. Cosine-similarity form revisits the conventional use of Gaussian latents with benefits from two critical properties: (1) the normalizing constant $Z_\gamma$ is independent of $\theta$ and can be omitted during training; (2) the scale hyperparameter $\gamma$ controls density magnitudes for training stability. Therefore, we adopt this form in our experiments.

### 3.1.3. STOCHASTIC AUGMENTATION STRATEGY

Training energy function via Eq.5 requires sampling negative samples from the marginal $p_\theta(\mathrm{x})$. To avoid long MCMC chains, we consider first generating initial samples through a generator, i.e., $\mathrm{x}^0 = G_\zeta(\mathrm{m})$, $\mathrm{m} \sim \mathcal{N}(0, I)$, then refining them with a few MCMC steps from marginal $E_\theta(\mathrm{x})$. However, this strategy exhibits a practical limitation: the initial sample distribution progressively gets closer to the data distribution during training, resulting in the energy function's catastrophic forgetting of low-density regions and earlier discovered modes. To mitigate this problem, we implement a stochastic augmentation strategy for negative samples before MCMC sampling.

$$\mathrm{x}_{\mathrm{aug}} = \begin{cases} v(\mathrm{x}^0), v \sim \mathcal{V} & \text{with probability } p, \\ \mathrm{x}^0, & \text{otherwise.} \end{cases} \qquad (13)$$

Each negative sample undergoes augmentation with Bernoulli probability $p$, where the augmented transformation $v \sim \mathcal{V}$ follows the random augmentations used in SSRL. This augmentation technique enables broader exploration of the energy landscape during training, facilitating diversity of MCMC chains. Empirically, this augmentation technique enhances OOD detection for distant outliers with minimal impact on generation quality.

## 3.2. Generator Training

We build a cooperative training framework where a generator initializes MCMC chains through one forward pass. Beyond the previous protocol (Xie et al., 2020), our framework features a joint energy function and a semantic-aware latent encoder, enabling us to investigate different generator training schemes through empirical analysis.

### 3.2.1. ENERGY DISTRIBUTION MATCHING (EM)

First, following traditional cooperative training, the generator can be optimized by minimizing the KL divergence between two joint distributions, i.e., $\min \mathrm{KL}\left(p_\theta(\mathrm{x}, \mathrm{m}) \| p_g(\mathrm{x}, \mathrm{m})\right)$, both distributions built from a Gaussian prior $p(\mathrm{m})$ and conditional $p(\mathrm{x}|\mathrm{m})$. Under the assumption that $p_g(\mathrm{x}|\mathrm{m})$ follows a Gaussian distribution, the objective simplifies to an MSE loss:

$$L_G = \frac{1}{n} \sum_{i=1}^{n} \frac{1}{2\tau^2} \left\| G_\zeta(\mathrm{m}_i) - \tilde{\mathrm{x}}_i^T \right\|_2^2, \tag{14}$$

where $i$ denotes the $i^{th}$ number of a batch with size $n$. $\tau^2$ is the fixed variance of $p_g(\mathrm{x}|\mathrm{m})$. $\tilde{\mathrm{x}}_i^T$ is the refined samples by running $T$ steps of MCMC from initial point $\tilde{\mathrm{x}}_i^0 = G_\zeta(\mathrm{m}_i)$:

$$\tilde{\mathrm{x}}_i^{t+1} = \tilde{\mathrm{x}}_i^t - \frac{\delta^2}{2} \nabla_{\tilde{\mathrm{x}}} E_\theta(\tilde{\mathrm{x}}_i^t) + \delta \epsilon^t, \quad \epsilon^t \sim \mathcal{N}(0, I) \tag{15}$$

$E_\theta$ is the marginal energy defined in Proposition 3.1. We empirically observe that this marginal energy MCMC performs well across all evaluated datasets.

We also investigate MCMC refinement from the perspective of the joint energy function. We take the basic idea of auxiliary variable MCMC (Brooks et al., 2011; Song & Ou, 2018) to sample in the augmented space $(\mathrm{x}, \mathrm{z})$. To circumvent the computational burden of two Markov chains in both data and latent spaces, we employ our latent encoder to perform a single MCMC procedure. Specifically, we first sample initial $\tilde{\mathrm{x}}_i^0 = G_\zeta(\mathrm{m}_i)$, followed by executing MCMC as described below:

$$\tilde{\mathrm{x}}_i^{t+1} = \tilde{\mathrm{x}}_i^t - \frac{\delta^2}{2} \nabla_{\tilde{\mathrm{x}}} E_\theta(\tilde{\mathrm{x}}_i^t, h(\tilde{\mathrm{x}}_i^t)) + \delta \epsilon^t, \quad \epsilon^t \sim \mathcal{N}(0, I) \tag{16}$$

This procedure[1] constrains $\tilde{\mathrm{x}}$ within the latent space, reducing the search space and improving efficiency. We observe that this joint energy refinement accelerates training in the early stage, but ultimately underperforms marginal MCMC when dealing with multimodal distributions.

### 3.2.2. ENERGY AND REAL DISTRIBUTION MATCHING (ERM)

Energy distribution matching aligns only the energy distributions, without direct access to the training data. Inspired by EC-VAE (Luo et al., 2024), also leveraging our latent encoder, we optimize the generator to match both the energy and real data distribution:

$$\begin{aligned} L_G &= \omega_1 \, \mathrm{KL}(p_\theta(\mathrm{x}, \mathrm{m}) \| p_g(\mathrm{x}, \mathrm{m})) \\ &\quad + \omega_2 \, \mathrm{KL}(p_{\mathrm{data}}(\mathrm{x}, \mathrm{z}) \| p_g(\mathrm{x}, \mathrm{z})), \end{aligned} \tag{17}$$

where $\omega_1$ and $\omega_2$ denote the importance weighting between two divergence components. The first term is equal to Eq.14. For the second term, we define $p_g(\mathrm{x}, \mathrm{z})$ as:

$$p_g(\mathrm{x}, \mathrm{z}) = \int p(\mathrm{m}) p_g(\mathrm{x}, \mathrm{z}|\mathrm{m}) d\mathrm{m}, \tag{18}$$

where $p_g(\mathrm{x}, \mathrm{z}|\mathrm{m})$ is defined as:

$$p_g(\mathrm{x}, \mathrm{z}|\mathrm{m}) = p_g(\mathrm{x}|\mathrm{m}) \frac{\exp(\rho \sin(\mathrm{z}, L(G_\zeta(\mathrm{m}))))}{Z_\rho}, \tag{19}$$

$L(G_\zeta(\mathrm{m}))$ denotes a latent mapping function of $G_\zeta(\mathrm{m})$, which can take either $h(G_\zeta(\mathrm{m}))$ or $g_{\psi^-}\left(f_{\phi^-}(G_\zeta(\mathrm{m}))\right)$, corresponding to the latent encoder and energy posterior, respectively. Here, $\psi^-$ and $\phi^-$ denote stop-gradient parameters. Both choices constrain latent alignment as well as pixel-level fidelity. Then the second term in Eq.17 can be optimized by the classic evidence lower bound (ELBO):

$$\mathbb{E}_{p_{\mathrm{data}}(\mathrm{x}, \mathrm{z})} \mathbb{E}_{q_\alpha(\mathrm{m}|\mathrm{x})} \left[ \log p_g(\mathrm{x}, \mathrm{z}|\mathrm{m}) - \log \frac{q_\alpha(\mathrm{m}|\mathrm{x})}{p(\mathrm{m})} \right] \tag{20}$$

where $q_\alpha$ denotes an inference model parameterized by $\alpha$ and jointly trained with the generator. This method introduces an extra network, while increasing training complexity, this autoencoder-based architecture would be necessary for applications such as image restoration.

## 4. Experiments

We conduct comprehensive experiments to evaluate our proposed method under various scenarios, including unconditional image generation, OOD detection, conditional sampling, and zero-shot image restoration. For our pretrained latent encoder, we choose SimCLR[2] (Chen et al., 2020) as our normalized SSRL method (detailed in section K).

---

[1]It can be viewed as refining $(\mathrm{x}, h(\mathrm{x}))$ using $E_\theta(\mathrm{x}, \mathrm{z})$. This is justified since $p_\theta(\mathrm{z}|\mathrm{x})$ is trained to match $p_{\mathrm{data}}(\mathrm{z}|\mathrm{x})$, and $h(\mathrm{x})$ can be viewed as approximate samples from $p_{\mathrm{data}}(\mathrm{z}|\mathrm{x})$.

[2]We implement SimCLR using the official code of W-MSE: https://github.com/htdt/self-supervised

*Table 1.* Generative performance on CIFAR-10. "w/o MCMC" denotes direct sampling from the generator without energy-based refinement via MCMC sampling. † indicates the modified model, same as above.

| Model | NFE↓ | FID↓ | IS↑ | | Model | NFE↓ | FID↓ | IS↑ |
|---|---|---|---|---|---|---|---|---|
| **Likelihood-based** | | | | | **EBM-based** | | | |
| PixelCNN (Oord et al., 2016) | 1024 | 65.9 | 4.60 | | IGEBM (Du & Mordatch, 2019) | 60 | 38.2 | 6.78 |
| Glow (Kingma & Dhariwal, 2018) | 1 | 48.9 | 3.92 | | joint Triangle (Han et al., 2020) | 1 | 30.10 | 7.17 |
| VAE (Kingma & Welling, 2014) | 1 | 115.8 | 3.8 | | CoopNets (Xie et al., 2020) | 51 | 33.61 | 6.55 |
| NVAE (Vahdat & Kautz, 2020) | 1 | 51.67 | 5.51 | | EBMBB (Geng et al., 2021) | 1 | 28.63 | 7.45 |
| **GAN-based** | | | | | VAEBM (Xiao et al., 2021) | 16 | 12.19 | 8.43 |
| | | | | | DRL (Gao et al., 2021) | 180 | 9.58 | 8.30 |
| WGAN-GP (Gulrajani et al., 2017) | 1 | 36.4 | 6.50 | | Hat EBM (Hill et al., 2022) | 51 | 19.30 | – |
| SN-GAN (Miyato et al., 2018) | 1 | 21.7 | 8.22 | | CLEL-Large (Lee et al., 2023) | 1200 | 8.61 | – |
| BigGAN (Brock et al., 2019) | 1 | 14.73 | 9.22 | | Dual-MCMC (Cui & Han, 2023) | 31 | 9.26 | 8.55 |
| StyleGAN2 w/ ADA(Karras et al., 2020) | 1 | 2.92 | 9.83 | | DDAEBM (Geng et al., 2024) | 4 | 4.82 | 8.86 |
| DDGAN (Xiao et al., 2022) | 4 | 3.75 | 9.63 | | CDRL (Zhu et al., 2024) | 96 | 4.31 | 9.17 |
| ACT (Kong et al., 2024) | 1 | 6.0 | 9.15 | | EC-VAE (Luo et al., 2024) | 1 | 5.20 | – |
| **Diffusion-based** | | | | | Energy Matching (Balcerak et al., 2025) | 325 | 3.34 | 9.61 |
| | | | | | **Ours** | | | |
| NCSN-v2 (Song & Ermon, 2020) | 1000 | 10.87 | 8.40 | | | | | |
| DDPM (Ho et al., 2020) | 1000 | 3.17 | 9.46 | | LGCEBM-EM w/o MCMC | 1 | 4.83 | 9.57 |
| NCSN++ (Song et al., 2021) | 2000 | 2.20 | **9.89** | | LGCEBM-EM | 16 | 4.23 | **9.76** |
| EDM (Karras et al., 2022) | 35 | **2.04** | 9.84 | | LGCEBM-ERML w/o MCMC | 1 | 6.05 | 9.14 |
| Flow Matching (Lipman et al., 2023) | 142 | 6.35 | – | | LGCEBM-ERML | 16 | 4.90 | 9.35 |
| Consistency Models (Song et al., 2023) | 1 | 8.70 | 8.49 | | LGCEBM$^\dagger$-EM w/o MCMC | 1 | 3.48 | 9.90 |
| | | | | | LGCEBM$^\dagger$-EM | 16 | **3.28** | **9.93** |

*Table 2.* Generative performance on CelebA-HQ 256.

| Model | FID ↓ |
|---|---|
| GLOW (Kingma & Dhariwal, 2018) | 68.93 |
| NVAE (Vahdat & Kautz, 2020) | 45.11 |
| VQGAN (Esser et al., 2021) | 10.2 |
| DDGAN (Xiao et al., 2022) | 7.64 |
| Score SDE (Song et al., 2021) | 7.23 |
| **EBM-based** | |
| VAEBM (Xiao et al., 2021) | 20.38 |
| Dual MCMC (Cui & Han, 2023) | 15.89 |
| CDRL (Zhu et al., 2024) | 10.74 |
| EC-VAE (Luo et al., 2024) | 12.35 |
| LGCEBM-EM | 10.02 |
| LGCEBM-ERML | 8.65 |
| LGCEBM$^\dagger$-ERML | **6.99** |

*Table 3.* Generative performance on ImageNet 32.

| Model | FID ↓ |
|---|---|
| PixelCNN (Oord et al., 2016) | 40.51 |
| DDPM++ (Kim et al., 2021) | 8.42 |
| Flow Matching (Lipman et al., 2023) | 5.02 |
| **EBM-based** | |
| EBM-CD (Du et al., 2021) | 32.48 |
| CLEL-Large (Lee et al., 2023) | 15.47 |
| CDRL (Zhu et al., 2024) | 9.35 |
| EC-VAE (Luo et al., 2024) | 5.76 |
| EBM$_{\text{MI+diff}}$ (Geng et al., 2025) | 6.57 |
| Energy Matching (Balcerak et al., 2025) | 6.64 |
| LGCEBM-EM | **4.45** |
| LGCEBM-ERML | 5.01 |
| LGCEBM$^\dagger$-EM | **3.90** |

We adopt the same architecture as Dual-MCMC (Cui & Han, 2023) for our generator and inference model. We use the energy function backbone from Dual-MCMC as our $f_\phi$ in $E_\theta(\mathbf{x}, \mathbf{z})$, while our $g_\psi$ in $p_{\phi,\psi}(\mathbf{z}|\mathbf{x})$ is implemented as a two-layer MLP with ReLU activation and no Batch Normalization (Ioffe & Szegedy, 2015), i.e., $g_\psi(\mathbf{u}) = W_2 \sigma(W_1\mathbf{u} + b_1) + b_2$. For the inference model $q_\alpha$, we follow Dual-MCMC and use a simple convolutional network with Leaky ReLU activation. We apply Exponential Moving Average (EMA) with a decay rate of

0.9999 to improve generation quality. We denote our model with EM-based generator training using Eq.15 and 16 as LGCEBM-EM and LGCEBM-EJM, and the ERM-based as LGCEBM-ERML and LGCEBM-ERME, corresponding to latent encoder and energy posterior forms of $L(G_\zeta(\mathbf{m}))$ in $p_g(\mathbf{x}, \mathbf{z}|\mathbf{m})$, respectively. We provide an algorithm and a list of acronyms in App.E to improve the readability of our method. For our implementation and main experiments, energy training uses the separable form in Eq.10 to define the joint energy function, together with the cosine-similarity

posterior in Eq.12. For generator training, we use the EM and ERML settings (see section 4.5 for comparisons).

## 4.1. Unconditional Image Generation

We showcase our model's capabilities in unconditional image generation on standard datasets involving CIFAR-10 (Krizhevsky et al., 2009), ImageNet 32 (Deng et al., 2009), and CelebA-HQ 256 (Liu et al., 2015). We construct $\mathcal{V}$ for sampling $p_{\text{data}}(z|x)$ using the random augmentations in SimCLR. We adopt the cosine-similarity posterior for $E_\theta(x, z)$ due to its stability and superior performance. For quantitative results, we adopt the commonly used Fréchet inception distance (FID) and Inception Score (IS) to evaluate sample fidelity and the number of function evaluations (NFE) to evaluate sampling efficiency. We show qualitative results in Fig.3 and quantitative results in Tabs.1-3 [3]. Fig.2 shows FID *vs.* network parameters on CIFAR-10. Note that recent generative models often require careful tuning and specialized architectures to achieve strong performance, we adopt advanced architectures and refined optimizer settings to enhance generation quality, referred to as the modified model (see section F). Unless stated otherwise, our model denotes the base model without these modifications. The modified model is trained only using the better-performing base-model settings for each dataset.

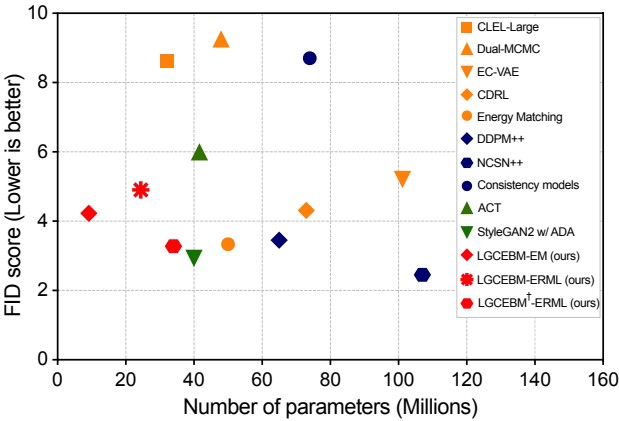

*Figure 2.* Param count vs. FID on CIFAR-10.

Our modified model achieves superior performance among EBMs, with fewer parameters and MCMC steps compared to strong baselines such as Energy Matching and CDRL. Even the base model significantly outperforms CLEL with much faster sampling, demonstrating the effectiveness of our cooperative training framework with an auxiliary generator. Our model also outperforms Dual-MCMC and EC-VAE by a large margin, validating that beyond cooperative training,

our latent-guided scheme can further improve generation. Notably, for single-step generation, our model surpasses strong diffusion baselines, including Consistency Model and its adversarial variant ACT. Moreover, our modified model achieves competitive performance with advanced GANs and Diffusion Models while using far fewer parameters. It gets the best IS score on CIFAR-10 and is the first EBM to beat Flow Matching on ImageNet 32.

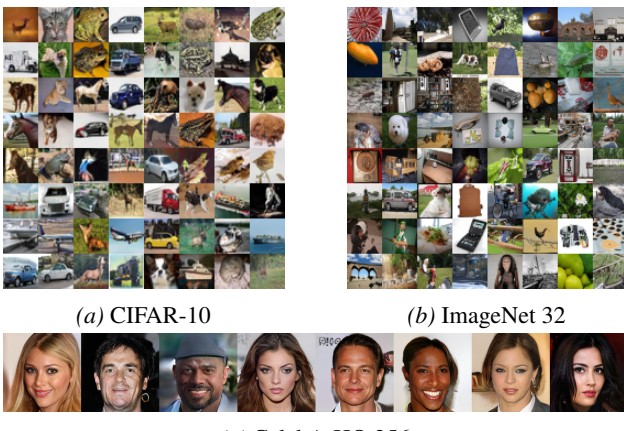

*(a)* CIFAR-10      *(b)* ImageNet 32

*(c)* CelebA-HQ 256

*Figure 3.* Samples generated by LGCEBM with MCMC refinement. Select models based on FID: EM for CIFAR-10/ImageNet-32; ERML for CelebA-HQ 256.

## 4.2. Out-of-distribution Detection

We evaluate our model's density modeling through OOD detection on CIFAR-10, using their unseen test sets as inliers and other datasets as outliers. We use the standard AUROC metric with a joint energy score inspired by CLEL:

$$s(x) := -F(f_\theta(x)) + \gamma \sim (g_\psi(f_\phi(x)), h(x)). \quad (21)$$

Results are shown in Tab.4. We observe that the joint energy score improves OOD detection on most datasets (with only slight degradation on SVHN), demonstrating enhanced robustness to diverse OOD samples through joint space modeling. Compared with other models, our model with joint energy score consistently performs at the top tier of likelihood-based models and matches specialized OOD methods. Notably, it shows significant improvement on CIFAR-100, which is challenging due to the similarity between CIFAR-100 and CIFAR-10. These show that semantic latent guidance substantially improves joint-space density modeling. Notably, our pretrained latent encoder is trained only on the in-distribution dataset without using large external data, so the OOD performance reflects the density modeling learned by the EBM. We reproduce Dual-MCMC and Hat EBM using their energy outputs as AUROC decision values. OOD results on ImageNet 32 are shown in App.H.

---

[3]Since baselines for ImageNet 32 and CelebA-HQ 256 are less established than CIFAR-10, we compare using FID and commonly reported baselines.

*Table 4.* AUROC with CIFAR-10 as in-distribution. $\left(-F\left(f_\theta(\mathbf{x})\right)\right)$ means $s(\mathbf{x}) := -F\left(f_\theta(\mathbf{x})\right)$ serves as the decision function.

| Method | SVHN | Constant | CIFAR-100 | CelebA |
|---|---|---|---|---|
| PixelCNN++ (Salimans et al., 2017) | 0.32 | 0.71 | 0.63 | – |
| GLOW (Kingma & Dhariwal, 2018) | 0.24 | – | 0.55 | 0.57 |
| NVAE (Vahdat & Kautz, 2020) | 0.44 | 0.65 | 0.49 | 0.68 |
| JEM (Duvenaud et al., 2020) | 0.67 | – | 0.67 | 0.75 |
| DRL (Gao et al., 2021) | 0.88 | 0.99 | 0.44 | 0.64 |
| Hat EBM (Hill et al., 2022) | 0.75 | 0.36 | 0.63 | 0.62 |
| CLEL (Lee et al., 2023) | 0.98 | – | 0.72 | 0.77 |
| Dual-MCMC(Cui & Han, 2023) | 0.62 | 0.32 | 0.54 | 0.59 |
| **Specialized OOD methods** | | | | |
| OOD EBM (Liu et al., 2020) | 0.91 | – | **0.87** | 0.78 |
| MPDR-S (Yoon et al., 2023) | **0.99** | **0.9996** | 0.56 | 0.73 |
| LGCEBM-EM $\left(-F\left(f_\theta(\mathbf{x})\right)\right)$ | 0.96 | 0.67 | 0.66 | 0.68 |
| LGCEBM-EM | 0.95 | 0.97 | 0.82 | 0.77 |
| LGCEBM-ERML $\left(-F\left(f_\theta(\mathbf{x})\right)\right)$ | 0.94 | 0.76 | 0.68 | 0.58 |
| LGCEBM-ERML | 0.95 | 0.96 | 0.82 | 0.75 |
| LGCEBM†-EM | 0.98 | 0.97 | 0.78 | **0.85** |

### 4.3. Conditional Sampling

We also investigate conditional sampling with our latent representation as labels. Our model uses a generator as an initializer, which offers fast sampling but requires the generator to produce high-quality initial samples. Therefore, similar to the ERM setting, we train an inference model to form an autoencoder with the generator under our EM framework, enabling us to obtain reconstructions from the input for initialization. We train our inference model using a variant of ELBO loss in the latent space to ensure detailed clarity and sharpness while preserving semantic similarity:

$$\mathbb{E}_{p_{\text{data}}(\mathbf{x},\mathbf{z})}\mathbb{E}_{q_\alpha(\mathbf{m}|\mathbf{x})}\left[\rho\operatorname{sim}(\mathbf{z}, h(G_\zeta(\mathbf{m}))) - \beta\log\frac{q_\alpha(\mathbf{m}|\mathbf{x})}{p(\mathbf{m})}\right]. \tag{22}$$

See App. I for details. We split our generation into two components $G(\mathbf{m}) + Y$. Following CLEL, we obtain the class representation $\overline{\mathbf{z_c}}$ for each class $c$, defined as the normalized average of latent representation across all images in class $c$. We draw an initialization $\overline{\mathbf{x_c}}$ as initial $G(\mathbf{m})$ by averaging all augmented images from each class. Then we iteratively optimize $Y$ and $\mathbf{m}$ by performing MCMC sampling from $E_\theta(G(\mathbf{m}) + Y, \overline{\mathbf{z_c}})$ and using the inference model conditioned on $G(\mathbf{m}) + Y$, respectively. From Fig.4 we can see that the EM setting is able to generate diverse samples with clear details for each class, whereas the ERML setting, while capable of generating some feature elements of the given class, fails to produce identifiable subjects. This is caused by the ELBO component in ERM training, which provides pixel-level reconstruction but produces blurry, low-sharpness results.

### 4.4. Image Restoration

We also present the application of our method in zero-shot image restoration tasks, including colorization and $8\times$ super-resolution. We conduct experiments on CelebA-HQ 256

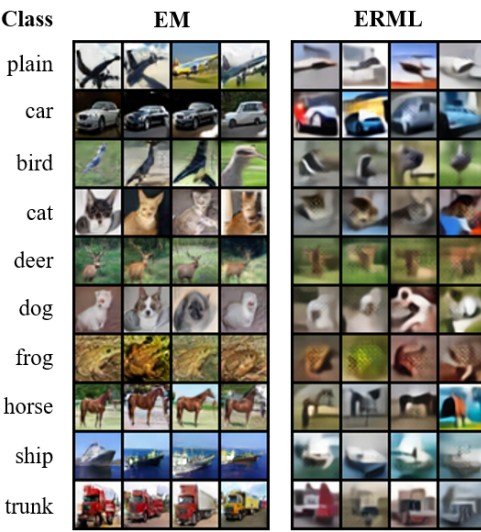

*Figure 4.* Conditional generated sample on CIFAR-10.

with ERML setting, since we need pixel-level restoration. Following Wang et al. (2023); Luo et al. (2024), we also use a linear operator $A$ to get the degraded image $y = \mathbf{A}\mathbf{x}$ and utilize its pseudo-inverse $\mathbf{A}^\dagger$ to derive the initial estimate $\hat{\mathbf{x}} = \mathbf{A}^\dagger y$. We obtain initial $\mathbf{m}^0$ using our inference model with input $\hat{\mathbf{x}}$. We use the following joint function to refine $\mathbf{m}$ via MCMC sampling:

$$p_{g,\theta}(\mathbf{A}^\dagger y, \mathbf{m}) \propto \exp\left(-E_\theta\left(G(\mathbf{m}), h\left(G(\mathbf{m})\right)\right)\right) \\ p(\mathbf{m})p\left(\mathbf{A}^\dagger y \mid \mathbf{A}^\dagger \mathbf{A} G(\mathbf{m})\right) \tag{23}$$

After refining $\mathbf{m}$, we update $Y$ using MCMC sampling with $G(\mathbf{m})$ fixed:

$$p_{g,\theta}(\mathbf{A}^\dagger y, Y) \propto \exp\left(-E_\theta\left(G(\mathbf{m}) + Y, h\left(G(\mathbf{m}) + Y\right)\right)\right) \\ p\left(\mathbf{A}^\dagger y \mid \mathbf{A}^\dagger \mathbf{A}\left(G(\mathbf{m}) + Y\right)\right) \tag{24}$$

We employ $\tilde{\mathbf{x}} = G(\mathbf{m}) + Y$ as our restoration solution. The qualitative results are shown in Fig.5 and the corresponding PSNR and SSIM metrics are reported in Tab.5. We can observe that with the help of joint energy distribution, our model can successfully restore those images with high quality and consistency after refinement on $\mathbf{m}$ and $Y$.

*Table 5.* Quantitative results of zero-shot image restoration on CelebA-HQ 256.

| Model | Colorization PSNR↑ / SSIM↑ | $8\times$ SR PSNR↑ / SSIM↑ |
|---|---|---|
| $G(\mathbf{m}^0)$ | 20.64 / 0.66 | 22.62 / 0.67 |
| $G(\mathbf{m})$ | 22.02 / 0.70 | 24.16 / 0.70 |
| $G(\mathbf{m}) + Y$ | 25.25 / 0.94 | 24.55 / 0.71 |

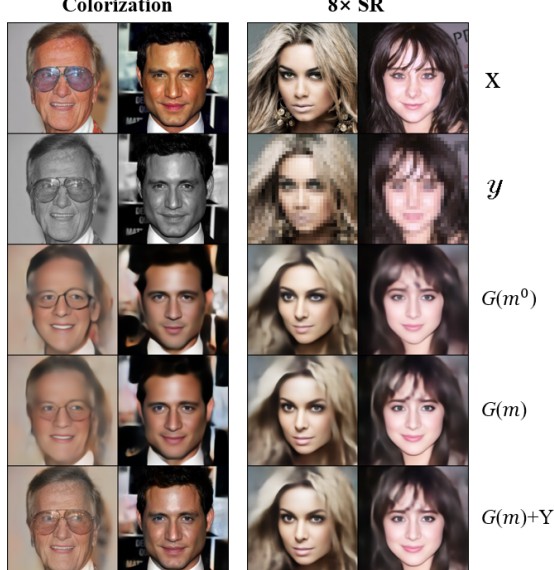

*Figure 5.* Qualitative results of zero-shot image restoration on CelebA-HQ 256.

*Table 6.* AUROC under different settings with CIFAR-10 as in-distribution.

| Method | SVHN | Constant | CIFAR-100 | CelebA |
|---|---|---|---|---|
| EM w/o pretrain | **0.97** | **0.997** | 0.75 | 0.71 |
| EM w/o Aug | 0.95 | 0.88 | **0.82** | 0.76 |
| EM | 0.95 | 0.97 | **0.82** | 0.77 |
| EJM | 0.95 | 0.94 | 0.81 | 0.73 |
| ERML | 0.95 | 0.96 | **0.82** | 0.75 |
| ERME | 0.95 | 0.98 | **0.82** | **0.78** |

## 4.5. Ablation Study

**Generality of $\mathcal{V}$ Choices and Energy Forms** Fig.6 compares FID with different $\mathcal{V}$ choices and energy forms. For $\mathcal{V}$ choices, we observe similar performance regardless of using random augmentations or uniform noise. This shows that our method is agnostic to the design of $\mathcal{V}$. For energy forms, while the Gaussian distribution is the conventional choice for modeling explicit posterior $p_{\phi,\psi}(z|x)$, we observe its sensitive variance effects. Thus, we constrain the log-variance to $[-1, 1]$ for the Gaussian posterior, and fix the variance for the conjugate exponential family form. Both perform better than the Gaussian-posterior Dual MCMC, yet remain inferior to the cosine-similarity posterior. Moreover, fixing the variance of the Gaussian posterior (denoted as 'Gaussian (fixvar)') yields comparable performance to the cosine-similarity form. This is expected, since the squared Euclidean distance satisfies $\|z_1 - z_2\|_2^2 = 2 - 2\sim(z_1, z_2)$, and the cosine-similarity form naturally induces a spherical Gaussian on the unit sphere. Therefore, we demonstrate that using a cosine-similarity posterior and decoupling the implicit data energy from the explicit posterior, as in Eq.10, offers greater flexibility and easier control.

**Impact of Component Configurations** Fig.7 tracks FID scores during training for the ablation study on CIFAR-10. Tab.6 shows their corresponding OOD performance. It can be seen that traditional EBM training without latent variables can not converge, no matter how the generator training is designed. Joint training with the latent encoder (denoted as "EM w/o pretrain"), as in CLEL, degrades both generation performance and OOD robustness. This phenomenon may stem from our generator-initialized EBM samples inadequately covering the true data manifold in the early stage of training, making their latent variables ineffective as negative representations for diversity. Augmentation technique can improve OOD results on Constant Dataset with negligible generation degradation. Generator training with EJM can get better results for the first stage, but finally slightly worse than EM setting (Eq.15). In particular, EJM tends to collapse towards the end of training on ImageNet 32. Hence, we recommend employing the EM setting. For the ERM setting, ERML and ERME achieve similar performance, indicating that our energy posterior tends to align with the representations learned by the latent encoder. We adopt ERML due to its fixed latent encoder.

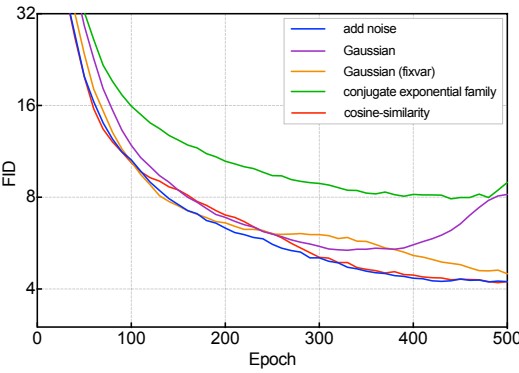

*Figure 6.* FID with different $\mathcal{V}$ choices and energy forms on CIFAR-10. "add noise" means adding uniform noise to construct $\mathcal{V}$. "Gaussian" and "cosine-similarity" mean two posterior choices.

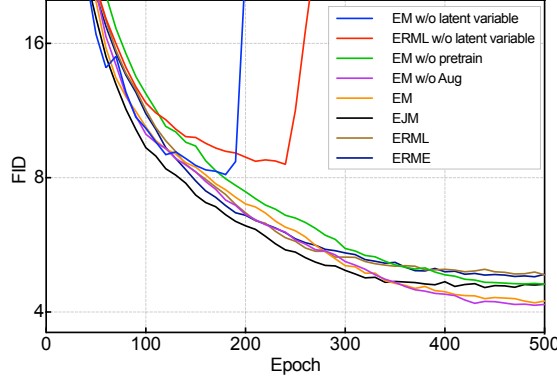

*Figure 7.* FID with different settings on CIFAR-10.

# 5. Conclusion

In this paper, we propose LGCEBM, a cooperative training scheme that jointly learns a latent-guided EBM and its generator initializer. We leverage pretrained self-supervised representations as our target latent variables to guide the energy function in capturing the semantic structure of the data manifold. Our model narrows the gap between EBMs and mainstream generative models with computationally efficient resources. It also excels in various downstream tasks, such as OOD detection, conditional sampling, and zero-shot image restoration. Additionally, our framework could be extended to multimodal large models (Han et al., 2023; Team et al., 2023; Hurst et al., 2024) by treating the joint space as a multimodal space and replacing SSRL methods with advanced modal-alignment techniques. We hope our work brings to light the profound potential of EBMs as mainstream generative models and stimulate active research in this area.

# Impact Statement

This paper presents work whose goal is to advance the field of Machine Learning. There are many potential societal consequences of our work, none of which we feel must be specifically highlighted here.

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

## A. Related Work

Energy-based models (EBMs) represent a powerful class of generative models that offer explicit unnormalized density estimation and architectural flexibility. Traditional EBM training relies on maximum likelihood estimation (MLE) with Markov Chain Monte Carlo (MCMC) sampling, particularly Langevin dynamics. However, noise-initialized Langevin dynamics often suffer from slow convergence and computational inefficiency (Song & Kingma, 2021). Several techniques have been proposed to alleviate the expensive MCMC, such as Persistent Contrastive Divergence (PCD) (Tieleman, 2008), adding a replay buffer (Du & Mordatch, 2019), short-run MCMC (Nijkamp et al., 2019), et.al. Nevertheless, these approaches remain inefficient as they still require hundreds to thousands of MCMC steps. Cooperative learning methods (Xie et al., 2020; 2021b; 2022; Hill et al., 2022) introduce a generator as a fast initializer learned to amortize long-run MCMC. Adversarial EBMs (Kumar et al., 2019; Geng et al., 2021; Grathwohl et al., 2021; Wan et al., 2025) form a minimax game between the energy function and the introduced generator to enable MCMC-free training. Some advances link connections between EBMs and other generative models to benefit from their strengths, such as VAE (Xiao et al., 2021; Luo et al., 2024), flow-based models (Gao et al., 2020; Nijkamp et al., 2022), and diffusion-based models (Gao et al., 2021; Zhu et al., 2024; Geng et al., 2024). Some recent works (Neklyudov et al., 2023; Loo et al., 2025; Balcerak et al., 2025) formulate EBMs from a vector field perspective, but slow sampling remains an issue.

Latent-variable EBMs define an energy function to characterize the joint density over data and latent variables. CLEL (Lee et al., 2023) leverages contrastive representation learning to learn meaningful latent structures that subsequently guide the EBM training. CEBM (Wu et al., 2021) decomposes the joint density into an intractable data distribution and a tractable latent posterior, providing VAE-like functionality while preserving EBM interpretability and density estimation. Divergence Triangle (Han et al., 2019; 2020) and Dual-MCMC (Cui & Han, 2023) build a unified framework that employs divergence triangle formulations to seamlessly integrate energy function, generator, and inference model through minimizing KL divergences between joint distributions. We focus on cooperative training between the generator and latent-guided EBM, decoupling the latent distribution from the generator's prior to retain informative latent representations.

## B. Preliminary of EBMs

Let $\mathcal{X}$ be the data space and $p_{\text{data}}(\mathrm{x})$ be true data distribution. An EBM defines a probability distribution through an energy function $E_\theta : \mathcal{X} \to \mathbb{R}$ parameterized by $\theta$,

$$p_\theta(\mathrm{x}) = \frac{\exp\left(-E_\theta(\mathrm{x})\right)}{Z_\theta}, \quad Z_\theta = \int \exp\left(-E_\theta(\mathrm{x})\right) d\mathrm{x}, \tag{25}$$

where $Z_\theta$ is the intractable normalizing constant. EBMs primarily rely on minimizing the negative log-likelihood for training such that:

$$L(\theta) := -\mathbb{E}_{\mathrm{x} \sim p_{\text{data}}(\mathrm{x})}\left[\log p_\theta(\mathrm{x})\right] = \mathbb{E}_{\mathrm{x} \sim p_{\text{data}}(\mathrm{x})}\left[E_\theta(\mathrm{x})\right] + \log Z_\theta. \tag{26}$$

The gradient of $L(\theta)$ can be derived as:

$$\frac{\partial L}{\partial \theta} = \mathbb{E}_{\mathrm{x} \sim p_{\text{data}}(\mathrm{x})}\left[\frac{\partial}{\partial \theta} E_\theta(\mathrm{x})\right] - \mathbb{E}_{\mathrm{x} \sim p_\theta(\mathrm{x})}\left[\frac{\partial}{\partial \theta} E_\theta(\mathrm{x})\right]. \tag{27}$$

Eq.27 requires MCMC sampling from energy distribution $p_\theta(\mathrm{x})$, which can be achieved by Langevin dynamics (Welling & Teh, 2011):

$$\mathrm{x}^{t+1} = \mathrm{x}^t - \frac{\delta^2}{2} \nabla_{\mathrm{x}} E_\theta(\mathrm{x}^t) + \delta \epsilon^t, \tag{28}$$

where $t$ indexes the time step, $\delta$ is the step size, and $\epsilon \sim \mathcal{N}(0, I)$. For small enough $\epsilon$ and large enough $t$, the distribution of $\mathrm{x}^t$ weakly converges to the energy distribution $p_\theta(\mathrm{x})$ regardless of the initial distribution of $\mathrm{x}^0$ (Neal, 2011; Raginsky et al., 2017; Xu et al., 2018).

## C. Proof of Proposition 3.1

Proof: From Eq.3, we have

$$\frac{\partial L}{\partial \theta} = \mathbb{E}_{(\mathrm{x},\mathrm{z}) \sim p_{\text{data}}(\mathrm{x},\mathrm{z})}\left[\frac{\partial}{\partial \theta} E_\theta(\mathrm{x}, \mathrm{z})\right] - \mathbb{E}_{(\mathrm{x},\mathrm{z}) \sim p_\theta(\mathrm{x},\mathrm{z})}\left[\frac{\partial}{\partial \theta} E_\theta(\mathrm{x}, \mathrm{z})\right] \tag{29}$$

Since $E_\theta(\mathrm{x}) = -\log \int \exp\left(-E_\theta(\mathrm{x}, \mathrm{z})\right) dz$, then $p_\theta(\mathrm{x}) = \int p_\theta(\mathrm{x}, \mathrm{z}) dz = \frac{\exp(-E_\theta(\mathrm{x}))}{Z_\theta}$, thus $E_\theta(\mathrm{x})$ is an available energy function of marginal $p_\theta(\mathrm{x})$. We can obtain:

$$p_\theta(\mathrm{x}, \mathrm{z}) = \frac{\exp\left(-E_\theta(\mathrm{x}, \mathrm{z})\right)}{Z_\theta} = \frac{\exp(-E_\theta(\mathrm{x}))}{Z_\theta} p_\theta(\mathrm{z}|\mathrm{x}) \tag{30}$$

$$\Rightarrow E_\theta(\mathrm{x}, \mathrm{z}) = E_\theta(\mathrm{x}) - \log p_\theta(\mathrm{z}|\mathrm{x}) \tag{31}$$

Substituting Eq.31 into the second term of Eq.29 yields:

$$
\begin{aligned}
\mathbb{E}_{(\mathrm{x}, \mathrm{z}) \sim p_\theta(\mathrm{x}, \mathrm{z})} \left[\frac{\partial}{\partial \theta} E_\theta(\mathrm{x}, \mathrm{z})\right] &= \mathbb{E}_{\mathrm{x} \sim p_\theta(\mathrm{x})} \left[\frac{\partial}{\partial \theta} E_\theta(\mathrm{x})\right] - \mathbb{E}_{(\mathrm{x}, \mathrm{z}) \sim p_\theta(\mathrm{x}, \mathrm{z})} \left[\frac{\partial}{\partial \theta} \log p_\theta(\mathrm{z}|\mathrm{x})\right] \\
&= \mathbb{E}_{\mathrm{x} \sim p_\theta(\mathrm{x})} \left[\frac{\partial}{\partial \theta} E_\theta(\mathrm{x})\right]
\end{aligned}
\tag{32}
$$

The second equality follows from:

$$\mathbb{E}_{(\mathrm{x}, \mathrm{z}) \sim p_\theta(\mathrm{x}, \mathrm{z})} \left[\frac{\partial}{\partial \theta} \log p_\theta(\mathrm{z}|\mathrm{x})\right] = \mathbb{E}_{\mathrm{x} \sim p_\theta(\mathrm{x})} \left[\int \frac{\partial}{\partial \theta} p_\theta(\mathrm{z}|\mathrm{x}) dz\right] = 0 \tag{33}$$

Plugging Eq.32 in Eq.29, we can get Eq.5.

## D. Conjugate Exponential Families

We express $E_\theta(\mathrm{x}, \mathrm{z})$ in the exponential family form (Wu et al., 2021),

$$E_\theta(\mathrm{x}, \mathrm{z}) = -\langle f_\phi(\mathrm{x}), \eta(\mathrm{z}) \rangle - \log p_\lambda(\mathrm{z}), \tag{34}$$

where $\log p_\lambda(\mathrm{z})$ is defined as a tractable exponential family and can be written as:

$$\log p(\mathrm{z}) = \langle \lambda, \eta(\mathrm{z}) \rangle - B(\lambda). \tag{35}$$

Then we can write $E_\theta(\mathrm{x}, \mathrm{z})$ as

$$E_\theta(\mathrm{x}, \mathrm{z}) = -\langle f_\phi(\mathrm{x}), \eta(\mathrm{z}) \rangle - (\langle \lambda, \eta(\mathrm{z}) \rangle - B(\lambda)) = -\langle \lambda + f_\phi(\mathrm{x}), \eta(\mathrm{z}) \rangle + B(\lambda). \tag{36}$$

Analogous to Eq.35, we derive the posterior

$$\log p_\theta(\mathrm{z}|\mathrm{x}) = \langle \lambda + f_\phi(\mathrm{x}), \eta(\mathrm{z}) \rangle - B(\lambda + f_\phi(\mathrm{x})). \tag{37}$$

For marginal energy $E_\theta(\mathrm{x})$, substituting Eqs.36 and 37 into Eq.31, we get

$$E_\theta(\mathrm{x}) = B(\lambda) - B\left(\lambda + f_\phi(\mathrm{x})\right) \tag{38}$$

## E. Training Procedure of LGCEBM

We provide the training procedure of our LGCEBM in algorithm 1 and a list of acronyms in Tab.7.

## F. Modified Model

To bridge the performance gap with other generative models, we adopt advanced architectures and carefully designed optimizer settings. For network architecture, we use the DDGAN (Xiao et al., 2022) discriminator as $f_\phi$ in $E_\theta(\mathrm{x}, \mathrm{z})$, with three modifications: (1) replacing the output scalar with a vector embedding, (2) changing input channels to three, and (3) removing time steps. $F$ is chosen as the base model. The generator follows the R3GAN architecture (Huang et al., 2024). All other components remain unchanged. Dataset augmentation and gradient penalties from R3GAN are also adopted to improve training stability and generation quality. We follow the R3GAN optimizer settings and apply a cosine schedule to the MCMC step size and $\gamma$ in Eq.12. For CIFAR-10 and CelebA-HQ, we use 15 MCMC steps with the step size of 0.01. For ImageNet 32, we use 5 MCMC steps with the step size of 1.0 for generator training and 0.01 for energy training. We additionally use a Softplus to the energy objective in Eq.5 to stabilize training. Our modified model achieves state-of-the-art performance among EBMs and outperforms DDGAN and R3GAN, whose network architectures we adopt, in FID with fewer parameters, as shown in Tab.8. Our model is comparable to GANs and Diffusion Models, demonstrating the strong generative capability of our latent-guided cooperative EBM. Further investigation of the modified training framework, including training schedules, architectures, and regularization, is left for future research.

---

**Algorithm 1** LGCEBM training

---

**Require:** a latent-variable EBM $E_\theta(x, z)$, a Generator $G_\zeta$, an inference model $q_\alpha$, a pretrained latent encoder $h$, an augmentation distribution $\mathcal{V}$, hyperparameters $\tau, p$.

1: **for** # training iterations **do**
2:     Sample $\{x_i\}_{i=1}^n \sim p_{\text{data}}(\mathbf{x})$.
3:     Sample $z_i = h(v(x_i)/\|h(v(x_i))\|_2, v \sim \mathcal{V}$ to get sample pairs $\{x_i, z_i\}_{i=1}^n \sim p_{\text{data}}(x, z)$
    ▷ Generator training
4:     Sample generated examples $\{G_\zeta(m_i)\}_{i=1}^n$, where $m_i \sim \mathcal{N}(0, I)$
5:     Obtain refined samples $\{\tilde{x}_i^T\}_{i=1}^n \sim p_\theta(\mathbf{x})$ initialized from $\tilde{x}_i^0 = G_\zeta(m_i)$ using Eq.15 or Eq.16.
6:     Compute $L_G = \frac{1}{n} \sum_{i=1}^n \frac{1}{2\tau^2} \left\| G(m_i) - \tilde{x}_i^T \right\|_2^2$
7:     **if** using ERM **then**
8:         Sample $\tilde{m}_i \sim q_\alpha(m|x_i)$ to get sample pairs $\{x_i, z_i, \tilde{m}_i\}_{i=1}^n$
9:         Compute $L_{ELBO} = \frac{1}{n} \sum_{i=1}^n \left[ \log p_g(x_i, z_i|\tilde{m}_i) - \text{KL}(q_\alpha(m|x_i)\|p(m)) \right]$ via Eq.19.
10:         Compute $L_G = L_G - L_{ELBO}$
11:     **end if**
12:     Update $G_\zeta, q_\alpha$ (if using ERM) by minimizing $L_G$
    ▷ EBM training
13:     Apply data augmentation to generated samples: $x_{\text{aug}_i} = v(G_\zeta(m_i)), v \sim \mathcal{V}$ with probability $p$ and $G_\zeta(m_i)$ otherwise
14:     Sample negative examples $\{\tilde{x}_i^T\}_{i=1}^n \sim p_\theta(\mathbf{x})$ initialized from $\tilde{x}_i^0 = x_{\text{aug}_i}$ using Eq.15.
15:     Compute $L_{\text{EBM}} \leftarrow \frac{1}{n} \sum_{i=1}^n E_\theta(x_i, z_i) - E_\theta(\tilde{x}_i^T)$
16:     Update $E_\theta$ by minimizing $\mathcal{L}_{\text{EBM}}$
17: **end for**

---

*Table 7.* List of acronyms used in this paper.

| Acronym | Meaning |
|---|---|
| EM | Training generator to match energy distribution with MCMC refinement under the marginal energy function using Eq.15 |
| EJM | Training generator to match energy distribution with MCMC refinement under the joint energy function using Eq.16 |
| ERM | Training generator to match both the energy and real data distribution using Eq.17 |
| ERML | ERM generator training with latent encoder form of $L(G_\zeta(m))$ in Eq.19 |
| ERME | ERM generator training with energy posterior form of $L(G_\zeta(m))$ in Eq.19 |
| w/o MCMC | Direct sampling from the generator without MCMC refinement |
| w/o latent variable | Traditional cooperative EBM training without latent variables |
| w/o pretrain | Joint training of the latent encoder and energy function |
| w/o Aug | Training without stochastic augmentation strategy |

*Table 8.* Comparison of DDGAN and R3GAN with our method.

| Model | FID↓ | IS↑ | NFE↓ | Params(M)↓ |
|---|---|---|---|---|
| DDGAN (Xiao et al., 2022) | 3.75 | 9.63 | 4 | 61 |
| R3GAN (Huang et al., 2024) | 3.64 | **10.00** | 1 | 41 |
| LGCEBM$^\dagger$-EM w/o MCMC | 3.48 | 9.90 | 1 | 34 |
| LGCEBM$^\dagger$-EM | **3.28** | 9.93 | 16 | 34 |

## G. Unconditional Generation on ImageNet 128

We evaluate our method on ImageNet 128, a high-dimensional and complex dataset. We use ImageNet 128 instead of the more common ImageNet 256 in Diffusion Models because ImageNet 128 is used in EBM comparisons, while ImageNet 256 has few EBM baselines. We compare our method with reported baselines whenever possible. We use DinoV2 (Oquab et al., 2023) for our pretrained latent encoder. Accordingly, we construct $\mathcal{V}$ by adding minor noise as in Eq.4. As shown in Tab.9, our method achieves superior results, outperforming Hat EBM while using the same architecture with 2048 channels. This demonstrates our method's potential for scaling to high-dimensional and complex data distributions.

*Table 9.* Unconditional generation on ImageNet 128.

| Model | FID↓ |
|---|---|
| SN-GAN (Miyato et al., 2018) | 65.7 |
| SS-GAN (Chen et al., 2019) | 43.9 |
| InfoMax GAN (Lee et al., 2021) | 58.9 |
| IGEBM (Du & Mordatch, 2019) | 43.7 |
| Hat EBM (Hill et al., 2022) | 29.37 |
| LGCEBM-EM | **25.70** |

## H. OOD Results on ImageNet 32

We also show our model's OOD performance on ImageNet 32 in Tab.10. Our model performs robustly on the challenging SVHN and Constant datasets, where likelihood-based methods such as VAE, GLOW, and PixelCNN typically fail. The joint energy score substantially improves OOD detection on FMNIST. Our modified model shows consistently strong performance across all datasets. Together, these results validate the enhanced density modeling capability of our latent-guided EBM.

*Table 10.* AUROC with ImageNet 32 as in-distribution. $\left(-F\left(f_\theta(\mathrm{x})\right)\right)$ means $s(\mathrm{x}) := -F\left(f_\theta(\mathrm{x})\right)$ serves as the decision function.

| Method | SVHN | Constant | FMNIST | CelebA |
|---|---|---|---|---|
| DAE (Vincent et al., 2008) | 0.10 | 0.07 | 0.991 | 0.43 |
| VAE (Kingma & Welling, 2014) | 0.13 | 0.03 | 0.95 | 0.55 |
| WAE (Tolstikhin et al., 2018) | 0.08 | 0.07 | 0.991 | 0.36 |
| PixelCNN++ (Salimans et al., 2017) | 0.03 | 0.00 | 0.004 | 0.24 |
| GLOW (Kingma & Dhariwal, 2018) | 0.17 | 0.41 | 0.86 | 0.48 |
| CLEL (Lee et al., 2023) | 0.96 | 0.83 | 0.54 | 0.74 |
| **Specialized OOD methods** | | | | |
| NAE (Yoon et al., 2021) | 0.985 | 0.97 | **0.994** | **0.95** |
| LGCEBM-EM $\left(-F\left(f_\theta(\mathrm{x})\right)\right)$ | 0.99 | 0.97 | 0.40 | 0.48 |
| LGCEBM-EM | 0.984 | 0.99 | 0.896 | 0.52 |
| LGCEBM-ERML $\left(-F\left(f_\theta(\mathrm{x})\right)\right)$ | 0.99 | 0.93 | 0.35 | 0.45 |
| LGCEBM-ERML | 0.985 | 0.99 | 0.868 | 0.54 |
| LGCEBM$^\dagger$-EM | **0.998** | **1.00** | 0.98 | 0.76 |

## I. Inference Model for EM Setting

We train an inference model for the EM setting using the following loss:

$$\mathbb{E}_{p_{\text{data}}(\mathrm{x},\mathrm{z})}\mathbb{E}_{q_\alpha(\mathrm{m}|\mathrm{x})}\left[\log p_g(\mathrm{z}|\mathrm{m}) - \beta \log \frac{q_\alpha(\mathrm{m}|\mathrm{x})}{p(\mathrm{m})}\right], \tag{39}$$

where $p_g(z|m) \propto \exp(\rho \, sim(z, L(G_\zeta(m))))$ aligns with Eq.19. The loss approximately minimizes the KL divergence between two conditional distributions, i.e., KL $(p_{data}(z|x)\|p_g(z|x))$ since:

$$\text{KL}\left(p_{data}(z|x)\|p_g(z|x)\right) = \text{KL}(p_{data}(x,z)\|p_g(x,z)) - \text{KL}(p_{data}(x)\|p_g(x)) \tag{40}$$

For the first term, is's equivalent to Eq.20, for the second term, it can be written as the classical ELBO:

$$\mathbb{E}_{p_{data}(x)}\mathbb{E}_{q_\alpha(m|x)}\left[\log p_g(x|m) - \log \frac{q_\alpha(m|x)}{p(m)}\right] \tag{41}$$

If we use the $\beta$-VAE ELBO in Eqs.20 and 41, Eq.40 can be approximately transformed into Eq.39. This approach ensures feature preservation in the latent space rather than enforcing pixel-level reconstruction. Fig.8 compares reconstruction results using Eq.39 versus the traditional ELBO loss in VAEs. The traditional ELBO fails to produce clear, semantically meaningful images with recognizable objects. Our training loss achieves high-quality reconstructions that preserve semantic properties of the input, such as object class, color, and visual style, without enforcing exact image reproduction. This indicates that our latent representation supports flexible instance generation.

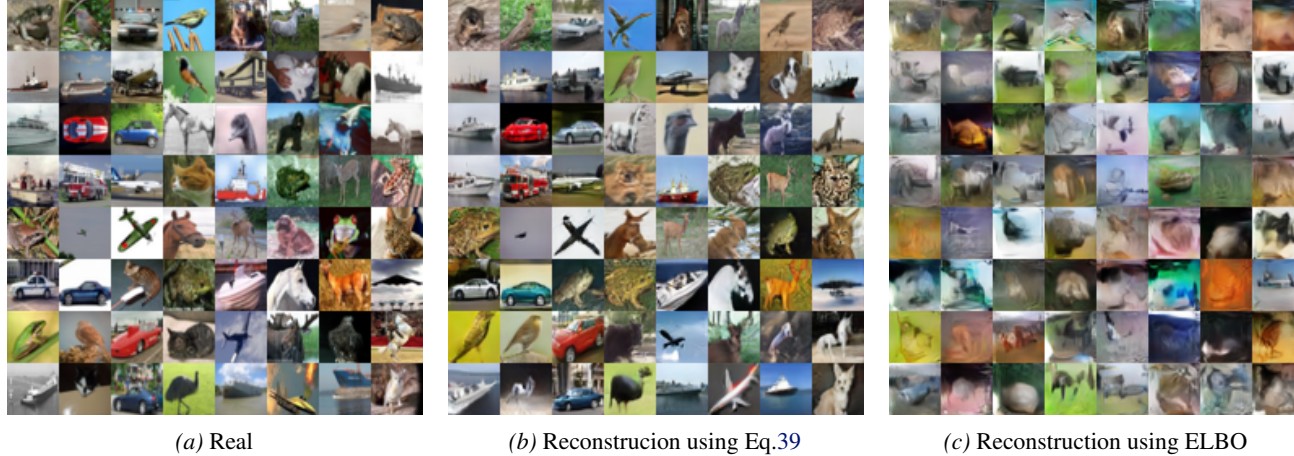

*(a)* Real        *(b)* Reconstrucion using Eq.39        *(c)* Reconstruction using ELBO

*Figure 8.* Reconstruction with different training losses.

## J. More Results of Image Restoration

We compare our method with strong image restoration baselines on CelebA-HQ, using FID for colorization and PSNR for $4\times$ super resolution. To be consistent with the baselines, we use $\mathbf{A}^\dagger y + (\mathbf{I} - \mathbf{A}^\dagger \mathbf{A})(G(m) + Y)$ as the restoration solution. As shown in Tab.11, our method achieves the best FID in colorization. For super resolution, we outperform the GAN-based PULSE and the EBM-based EC-VAE, and are competitive with the diffusion-based DDNM and DDRM. These results further confirm the effectiveness of our model for image restoration tasks.

*Table 11.* Comparison of image restoration on CelebA-HQ 256.

| **Model** | Colorization FID↓ | $4\times$ SR PSNR↑ |
|---|---|---|
| PULSE (Menon et al., 2020) | N/A | 22.7 |
| DDRM (Kawar et al., 2022) | 31.2 | **31.6** |
| DDNM (Wang et al., 2023) | 26.4 | **31.6** |
| EC-VAE (Luo et al., 2024) | 13.3 | 28.8 |
| LGCEBM-ERML | **12.9** | 30.8 |

## K. Adaptability to Various SSRL Methods

Our framework theoretically can be applied to any normalized self-supervised representation learning (SSRL) method. To verify our model's adaptability, we choose two other classic normalized SSRL methods, BYOL (Grill et al., 2020) and

W-MSE (Ermolov et al., 2021), to pretrain our latent encoder. Tab.12 reports FID and AUROC metrics for different SSRL methods. We can observe that SimCLR achieves the best FID performance among these three SSRL methods, and our LGCEBM scales well with various SSRL methods.

*Table 12.* Performance with different normalized SSRL methods.

| Method | FID↓ | AUROC↑ | | | |
|--------|------|--------|-----|-----|-----|
| | | SVHN | Constant | FMNIST | CelebA |
| BYOL | 5.14 | **0.96** | 0.98 | **0.85** | **0.81** |
| W-MSE | 5.07 | 0.93 | **0.99** | 0.83 | 0.77 |
| SimCLR | **4.23** | 0.95 | 0.97 | 0.82 | 0.77 |

## L. Reconstruction of LGCEBM-ERM

While our autoencoder-style ERM scheme is designed primarily for initialization, we additionally demonstrate its image reconstruction capabilities in Figs.9-10. Following the test setting in Han et al. (2019), we also compare our approach with other models that also incorporate an inferential mechanism, where performance is quantitatively measured by per-pixel mean square error (MSE). As shown in Tab.13, our model achieves the best performance on CIFAR-10, outperforming Dual-MCMC even with Langevin refinement. On CelebA 64, our model achieves comparable results to Dual-MCMC but without requiring additional Langevin dynamics.

*Table 13.* Reconstruction evaluation using MSE on CIFAR-10 and CelebA 64. Inf+L=10 denotes using 10-step Langevin dynamics initialized by the inference model.

| Methods | CIFAR-10 | CelebA 64 |
|---------|----------|-----------|
| WS (Hinton et al., 1995) | 0.058 | 0.152 |
| VAE (Kingma & Welling, 2014) | 0.037 | 0.039 |
| ALI (Dumoulin et al., 2016) | 0.311 | 0.519 |
| ALICE (Li et al., 2017) | 0.034 | 0.046 |
| Divergence Triangle (Han et al., 2019) | 0.028 | 0.030 |
| Dual-MCMC (Inf) (Cui & Han, 2023) | 0.049 | 0.022 |
| Dual-MCMC (Inf+L=10) (Cui & Han, 2023) | 0.024 | **0.013** |
| **LGCEBM-ERML** (Inf) | **0.019** | 0.014 |

## M. Hyperparameter Settings

We specify the hyperparameters used for our training on each dataset in Tab.14. We adopt two forms of function $F$ in Eq.10 for different datasets based on the generation performance. For CIFAR-10, ImageNet 32 and ImageNet 128, we define $F\left(f_\phi(\mathrm{x})\right) = \frac{\|f_\phi(\mathrm{x})\|_2^2}{2}$, while for CelebA 64 and CelebA-HQ 256, we define $F$ to be a learnable linear function, which is trained along with $E_\theta(\mathrm{x}, \mathrm{z})$. The output dimension of $f_\phi(\mathrm{x})$ is 512.

## N. Additional Results

We provide more qualitative visual results for both EM and ERM settings in Figs.11-13.

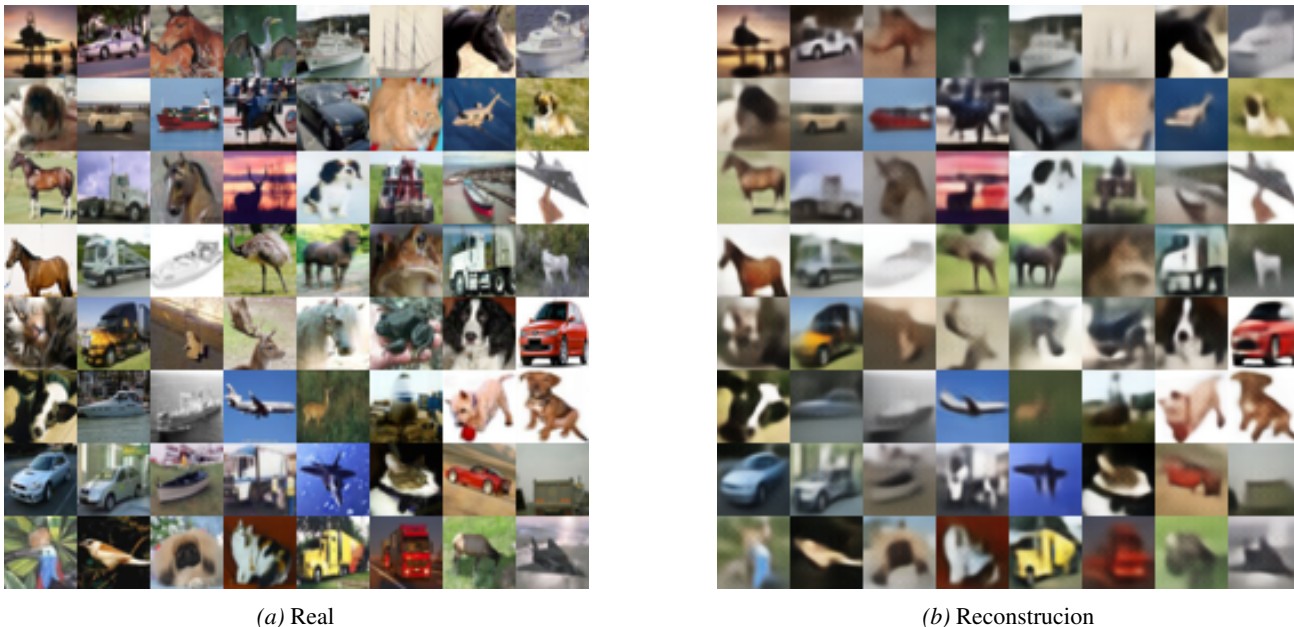

*(a)* Real                                        *(b)* Reconstrucion

*Figure 9.* Reconstruction of LGCEBM-ERML on CIFAR-10.

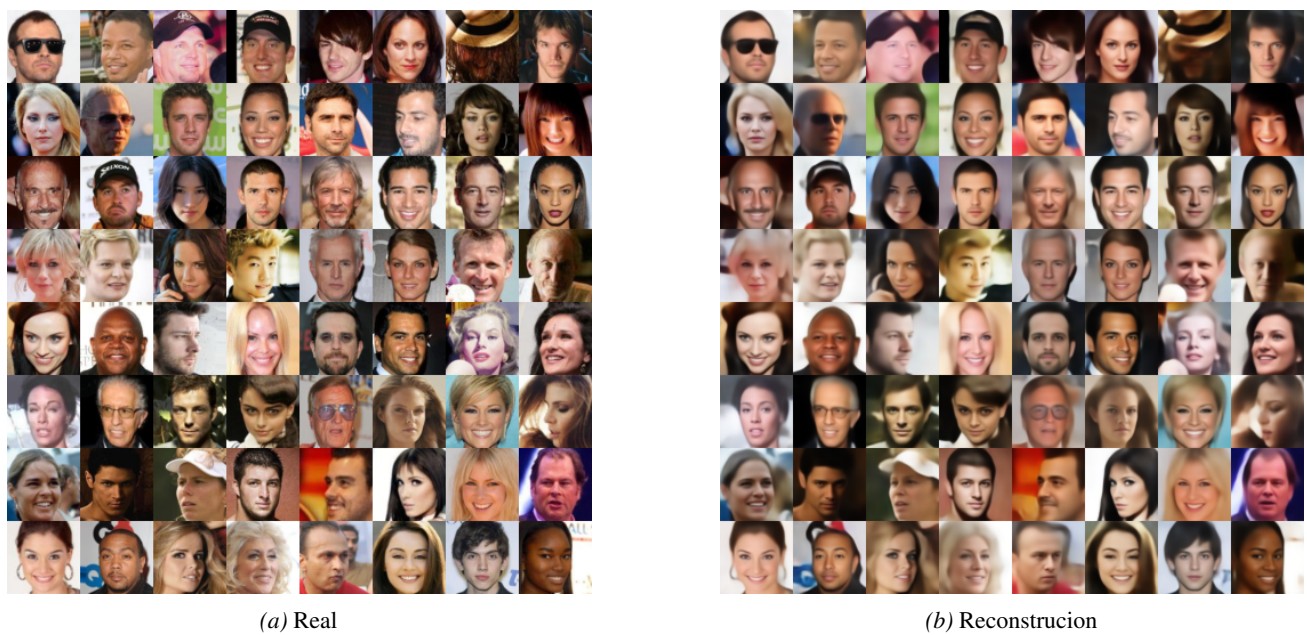

*(a)* Real                                        *(b)* Reconstrucion

*Figure 10.* Reconstruction of LGCEBM-ERML on CelebA 64.

*Table 14.* Hyperparameters for each dataset.

| | CIFAR-10 | ImageNet 32 | CelebA-HQ 256 |
|---|---|---|---|
| $E_\theta$ learning rate / Adam $\beta_1, \beta_2$ | 1e-4 / (0.0, 0.999) | 1e-4 / (0.0, 0.999) | 1e-4 / (0.0, 0.9) |
| $G$ learning rate / Adam $\beta_1, \beta_2$ | 2e-4 / (0.0, 0.9) | 2e-4 / (0.0, 0.9) | 3e-4 / (0.0, 0.9) |
| $q_\alpha$ learning rate / Adam $\beta_1, \beta_2$ | 2e-4 / (0.0, 0.9) | 2e-4 / (0.0, 0.9) | 1e-4 / (0.0, 0.9) |
| EMA decay rate | 0.9999 | 0.9999 | 0.9999 |
| $\gamma$ for training | 0.01 | 0.01 | 0.01 |
| $\gamma$ for OOD | 0.1 | 0.1 | 1 |
| batch size | 256 | 256 | 128 |
| MCMC steps | 15 | 15 | 15 |
| MCMC step size $\delta^2$ | 25 | 25 | 0.1 |
| $\omega_1$ / $\omega_2$ in Eq.17 | 1 / 0.1 | 1 / 0.1 | 70 / 1 |
| $\rho$ in Eq.19 | 1 | 1 | 50 |
| training epochs | 500 | 100 | 300 |
| data range | [0, 1] | [-1, 1] | [-1, 1] |
| latent dimension | 128 | 128 | 256 |
| $E_\theta, G$ hidden channels | 256 | 512 | 1024 |
| $q_\alpha$ hidden channels | 128 | 128 | 64 |
| $G$ params | 4.3M | 16.0M | 34.3M |
| $E_\theta$ params | 4.9M | 17.6M | 40.7M |
| $q_\alpha$ params | 15.2M | 15.2M | 8.1M |

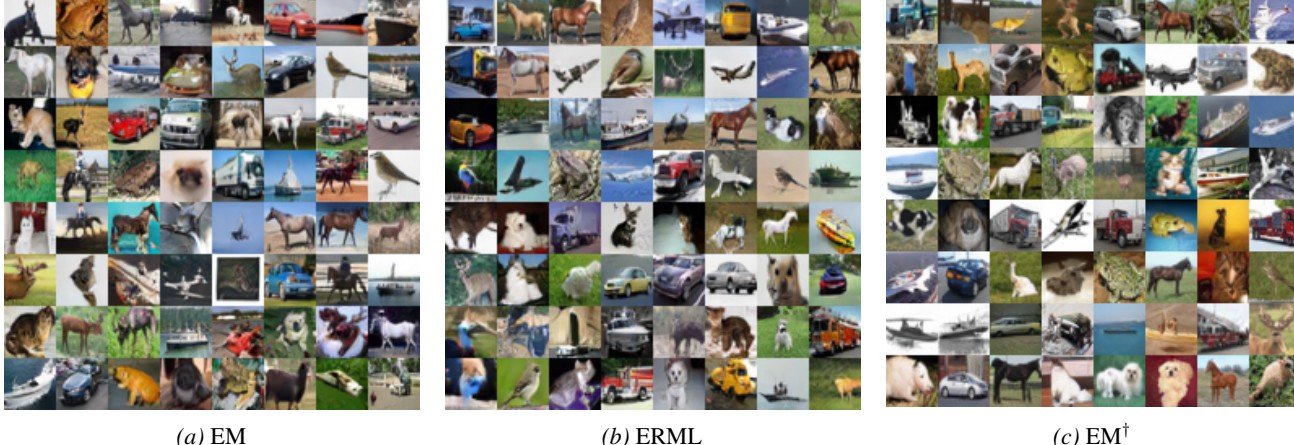

*(a)* EM        *(b)* ERML        *(c)* EM$^\dagger$

*Figure 11.* Samples generated by LGCEBM with MCMC refinement on CIFAR-10.

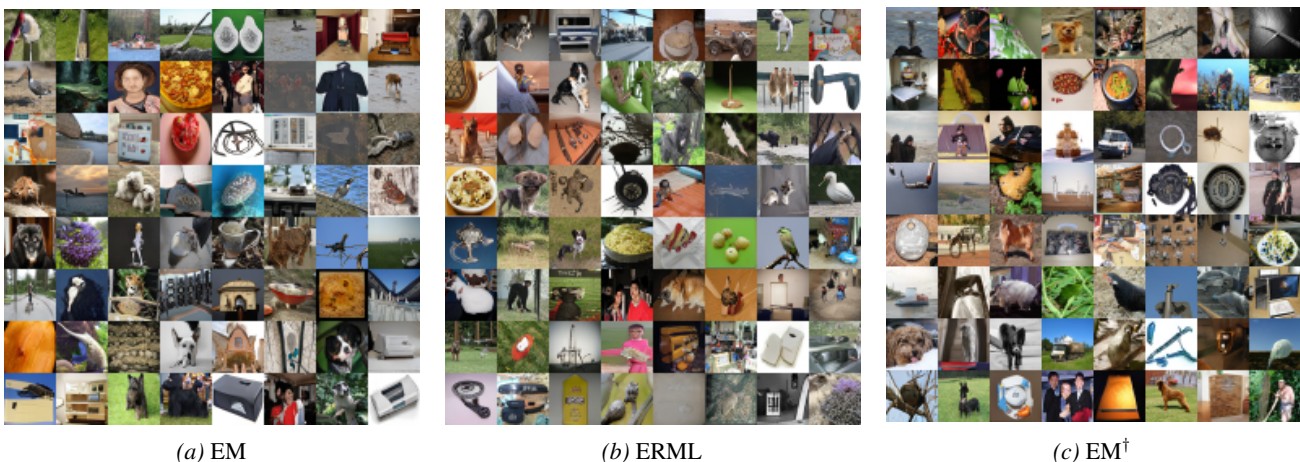

*(a)* EM           *(b)* ERML           *(c)* EM†

*Figure 12.* Samples generated by LGCEBM with MCMC refinement on ImageNet 32.

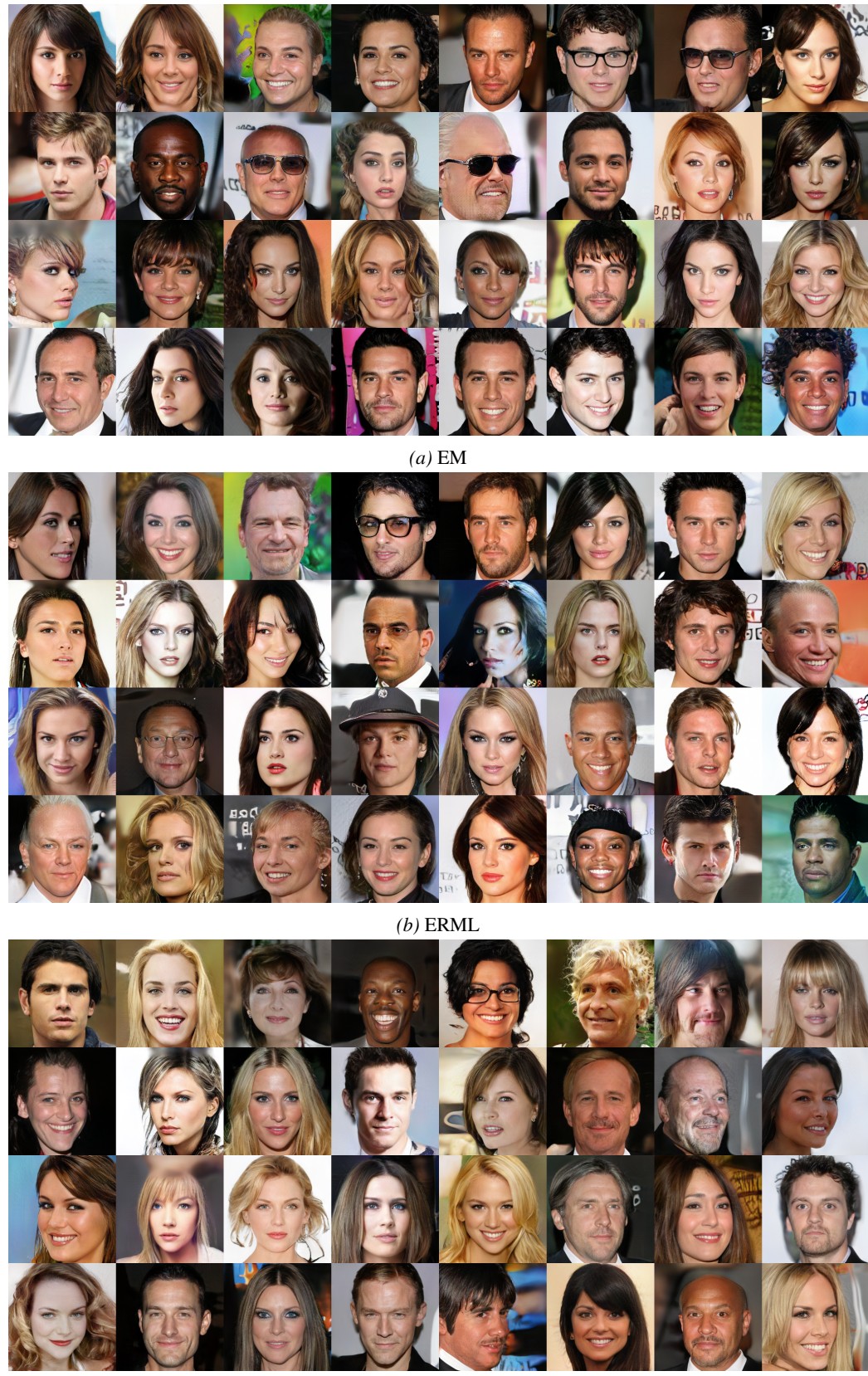

*(a)* EM

*(b)* ERML

*(c)* ERML[†]

*Figure 13.* Samples generated by LGCEBM with MCMC refinement on CelebA-HQ 256.

