# OpenReview forum: "Latent-Guided Cooperative Energy-Based Models"
_ICML.cc/2026/Conference — ICML 2026 regular_

### Official Review · Reviewer_j67t · 2026-03-08

**Soundness:** 3
**Presentation:** 3
**Significance:** 2
**Originality:** 2
**Overall Recommendation:** 4
**Confidence:** 3

**Summary:**

The paper introduces Latent-Guided Cooperative Energy-Based Models (LGCEBM), a novel framework designed to enhance both the training stability and generation quality of Energy-Based Models (EBMs). Its key innovation lies in leveraging a pre-trained self-supervised representation (e.g., SimCLR) to define a target joint distribution that captures both data density and semantic structure, thereby guiding the learning of the energy function. Experimental results show that the proposed method achieves competitive performance in out-of-distribution (OOD) detection, conditional sampling, and zero-shot image restoration.

**Compliance With Llm Reviewing Policy:**

Affirmed.

**Key Questions For Authors:**

1.To rigorously attribute this performance gain to the proposed latent-guided mechanism rather than architectural improvements, it is essential to compare against a standard Cooperative EBM without latent guidance or stochastic augmentation but trained with the exact same R3GAN/DDGAN backbone. Providing this comparison is crucial to validate the specific contribution of the latent-guided mechanism.
2. In Section 3.1.3, stochastic augmentation is applied to negative samples before MCMC initialization. However, modifying the initial states or transition dynamics of an MCMC chain can, in principle, alter its stationary distribution. Since the ultimate goal is to model the clean data distribution  $p_{data}(x)$ does training the energy function on augmented samples introduce a bias toward the augmented distribution?
3. There is a large FID gap between w/o MCMC (4.83) and Refined (3.28). Since the generator is trained to minimize the KL divergence with the energy model (Eq. 14), does this gap imply that the generator (with a simple Gaussian prior) inherently lacks the capacity to fit the complex, multi-modal distribution described by the energy function?

**Limitations:**

While inference time and parameter counts are provided, a discussion on the total computational cost (including the time required for SSL pre-training + EBM training + Inference model training) compared to single-stage baselines is missing.

**Strengths And Weaknesses:**

Strengths:
1.This result establishes a new Energy-Based Model (EBM), significantly outperforming existing baselines such as CLEL and Energy Matching, and substantially narrowing the performance gap with dominant generative paradigms like GANs and diffusion models.
2. The paper offers a valuable theoretical contribution: by rigorously justifying the optimization of the marginal energy as a means to learn the target joint distribution, the authors provide a solid foundation for circumventing MCMC sampling in the joint space. This insight renders the training of latent-variable EBMs markedly more efficient.
3.the work demonstrates broader applicability through tasks such as out-of-distribution (OOD) detection—conducted under fair in-distribution pre-training—and zero-shot image restoration. These results indicate that the learned energy function captures meaningful geometric and semantic structure, rather than merely modeling superficial data statistics.

Weaknesses:
1.The headline performance is achieved using a Modified Model built upon advanced R3GAN generator and DDGAN discriminator backbones. However, the paper does not provide a direct ablation study comparing this configuration against a standard Cooperative EBM trained on the exact same backbone. Without such a controlled comparison, it remains unclear how much of the improvement stems from the proposed Latent-Guided mechanism versus architectural upgrades. This makes it difficult to disentangle methodological contributions from backbone-driven gains.
2.Although the paper claims efficient sampling, Table 1 shows a substantial performance gap between the generator’s direct output (w/o MCMC: 4.83) and the refined output (Refined: 3.28). This discrepancy suggests that the generator does not fully distill the target energy distribution and still depends heavily on computationally expensive MCMC refinement to achieve state-of-the-art results. Consequently, the claimed efficiency advantages require further justification and quantitative analysis of the cost–performance trade-off.
3.The overall framework involves a multi-stage and relatively complex training pipeline, including:(1) pre-training an SSL encoder,(2) alternating optimization between the generator and EBM, and(3) training an additional inference model for the ERM setting. The core contribution lies in a principled integration and refinement of existing ideas (latent-variable modeling (e.g., CLEL), cooperative learning, and standard SSL pre-training) rather than the introduction of an entirely new learning paradigm. That said, the combination is non-trivial and yields empirically meaningful improvements, which constitutes a valuable advance in its own right.

---

> ### Author Rebuttal · Authors · 2026-03-31
>
> Thanks for your valuable review. We address your concerns below.
> ## Comparison between Cooperative EBM and Ours
> Thanks for raising this important and rigorous point. First, as illustrated in Fig.7, EM without latent variables reduces to cooperative EBM, which fails to converge. We also switched the generator training to our ERML setting, but it still could not achieve convergence. This suggests that our proposed latent-guided mechanism is essential for stabilizing training. Furthermore, we also evaluated cooperative EBM under the modified structure as you suggested. The results are shown below. We find that cooperative EBM performs worse than DDGAN and R3GAN, whereas with our latent guidance, LGCEBM outperforms both DDGAN and R3GAN. We believe these results sufficiently demonstrate the advantage of our latent-guided mechanism.
> | Model              | FID↓ | NFE↓ | Params(M)↓ |
> |--------------------|------|------|------------|
> | DDGAN   | 3.75  | 4    | 61         |
> | R3GAN  | 3.64 | 1    | 41         |
> | Cooperative EBM w/o MCMC         | 4.69  | 1    | 34         |
> | Cooperative EBM      | 4.41 | 1    | 34         |
> | LGCEBM†-EM w/o MCMC         | 3.48 | 1    | 34         |
> | LGCEBM†-EM                  | 3.28  | 16   | 34         |
>
> ## Distributional Bias Induced by Stochastic Augmentation
> Thanks for your insightful question! Our stochastic augmentation doesn't affect the positive term in energy loss and is applied before MCMC refinement, meaning that, if the subsequent MCMC process is properly configured[1,2], it should theoretically not introduce distributional bias toward the augmented distribution. From a practical perspective, we keep the augmentation probability low&thinsp;(no more than 0.05), so the effect of stochastic augmentation remains limited. And we assign high energy from augmented samples, helping shape regions away from the real data.
> Our OOD detection results&thinsp;(Tabs.4,6,9) demonstrate that stochastic augmentation does not impair energy modeling of the real data distribution.
>
>   ## FID Gap between w/o MCMC and Refined Samples
> There may be a slight misunderstanding regarding this point. Our refined result(3.28) is refined from the modified model(w/o MCMC: 3.48), as reported in Tab.7 of Appendix F&thinsp;(you can also see the table above). This single-step generation w/o MCMC already outperforms DDGAN and R3GAN. We will clarify this in the final paper.
>
> ## Training Cost
> We have shown the training cost below for comparison.
> | Method            | Memory usage(GB)↓ | Training iterations(K)↓ | FID↓ |
> |-------------------|------------------:|------------------------:|-----:|
> | R3GAN          | 57               | 195                   | 3.64 |
> | VAEBM             | 129               | 25                      | 12.2 |
> | CLEL              | 10                | 100                     | 8.61 |
> | CDRL              | 69                | 400                     | 4.31 |
> | LGCEBM-EM (ours)   | 14                | 100                     | 4.23 |
> | LGCEBM-ERML (ours)   | 16                | 100                     | 4.90 |
> | LGCEBM$^\dagger$-EM (ours)   | 24                | 100                     | 3.28 |
>
> The inference model and EBM are trained together. For SSRL pre-training, the trained encoder is available from several open-source resources and is widely adopted directly for generative modeling, such as REPA[3], RAE[4], and drifting model[5].
>
>
> [1] Nijkamp, Erik, et al. "On the anatomy of mcmc-based maximum likelihood learning of energy-based models." AAAI 2020.
>
> [2] Abdulsalam, Azwar, and Joseph G. Makin. "Revisiting Contrastive Divergence for Density Estimation and Sample Generation." TMLR 2025.
>
> [3] Yu, Sihyun, et al. "Representation alignment for generation: Training diffusion transformers is easier than you think." ICLR 2025.
>
> [4] Zheng, Boyang, et al. "Diffusion transformers with representation autoencoders." ICLR 2026.
>
> [5] Deng, Mingyang, et al. "Generative Modeling via Drifting." arXiv preprint arXiv:2602.04770.

---

> > ### Author Rebuttal · Reviewer_j67t · 2026-04-05
> >
> > Thanks for your reply.
> >
> > The rebuttal addresses my main concern about disentangling the contribution of the latent-guided mechanism from backbone improvements, and it also clarifies the apparent FID gap. Overall, these responses increase my confidence in the paper, and I therefore maintain my Weak Accept recommendation.

---

> > > ### Author Response · Authors · 2026-04-07
> > >
> > > Thanks for your acknowledgement! Your comments are instrumental in shaping the quality of our paper. We further compare OOD detection performance alongside generation quality between Cooperative EBM and ours to demonstrate the significant advantages of our proposed approach.
> > > | Method | AUROC↑ |  |  |  | Generation |  |
> > > |---|---|---|---|---|---|---|
> > > |  | SVHN | Constant | CIFAR-100 | CelebA | FID↓ | NFE↓ |
> > > | Cooperative EBM | 0.62 | 0.82    | 0.52     | 0.66   | 4.41 | 16 |
> > > | LGCEBM†-EM | 0.98 | 0.97 | 0.78      | 0.85   | 3.28 | 16 |
> > >
> > > Our original manuscript and rebuttal response demonstrate the superiority of our method from multiple perspectives, such as generation quality and OOD detection performance, greatly narrowing the gap between EBMs and mainstream generative models. Our latent-guided mechanism helps the energy function capture both the semantic geometry of the data manifold and the data likelihood. Modeling data distributions together with their geometric structure is the foundational motivation and the ultimate objective of generative models, while most existing methods mainly focus on the former. Thus, as a parting remark, we would be grateful if you would consider raising your score in light of our method's potential to advance EBM research and the broader generative modeling.

---

### Official Review · Reviewer_ViZV · 2026-03-11

**Soundness:** 3
**Presentation:** 2
**Significance:** 3
**Originality:** 3
**Overall Recommendation:** 4
**Confidence:** 2

**Summary:**

The paper proposes Latent-Guided Cooperative Energy-Based Models (LGCEBM). It tries to improve the training stability and generation quality of EBMs. The method relies on a pretrained encoder to provide target latent variables. The authors define a joint energy function. They show that MCMC sampling is only required in the data space. An auxiliary generator is trained jointly to initialize the MCMC chains. The method is evaluated on unconditional image generation, OOD detection, conditional sampling, and zero-shot image restoration.

**Compliance With Llm Reviewing Policy:**

Affirmed.

**Key Questions For Authors:**

1. Is there a notation inconsistent? Eq. 14 uses $\overline{x}_i^T$ while Eq. 15 uses $\tilde{x}_i^T$.
2. In Appendix F, you mention applying a cosine schedule to $\gamma$. Could the authors clarify why the normalizing constant $Z$ can be omitted when $\gamma$ varies during training?
3. How does the EBM perform on OOD detection if you freeze the  model but use a randomly initialized latent encoder? I want to isolate the density modeling capability of the EBM from the power of SimCLR.

**Limitations:**

The authors could analyze deeper why ERML method suffers from mode collapse and blurry outputs.

**Strengths And Weaknesses:**

Strengths:

- The mathematical simplification in proposition 3.1 is elegant. It successfully avoids expensive joint space MCMC sampling.
- The stochastic augmentation strategy for negative samples is simple but highly effective.
- Empirical performance on CIFAR-10 and ImageNet 32 is  competitive for the EBM family.

Weaknesses:

- SimCLR representations inherently cluster in-distribution data and isolate OOD data. The high AUROC likely stems directly from the pretrained SimCLR encoder rather than the EBM's learned density.
- There is a mathematical inconsistency regarding Eq. 12. The text claims the normalizing constant $Z$ is independent of $\theta$ and can be omitted. This is only true if $\gamma$ is a fixed constant in my understanding. However, Appendix F states a cosine schedule is applied to $\gamma$. If $\gamma$ changes over time, $Z(\gamma)$ changes.

---

> ### Author Rebuttal · Authors · 2026-03-31
>
> Thanks for your valuable review. We address your concerns below.
> ## Notation Clarification
> We both use $\tilde{x}_i^T$ in Eqs.14 and 15. Please check them again.
> ## Omitting the Normalizing Constant $Z_\gamma$ during Training
> The normalizing constant $Z_\gamma$ depends only on $\gamma$, which is not a trainable parameter. Our trainable parameters for the energy function are $\theta=(\phi,\psi)$, and $Z_\gamma$ is independent of them. Therefore, $Z_\gamma$ can be omitted regardless of whether $\gamma$ varies. In fact, we vary $\gamma$ only in our modified model, and its effect on performance is small.
> ## Source of the OOD Performance Improvement
> Thanks for your insightful and thought-provoking question! Our OOD performance improvement stems from our latent-guided mechanism, but the gain should not be attributed solely to SimCLR. As presented in Tab.11 of Appendix K, we also evaluated other SSRL methods, some of which even outperformed SimCLR. Secondly, we vary $\gamma$ in our joint energy score(Eq.21) to examine how much density modeling capability each term contributes. As shown below, we find that the latent-guided term improves OOD performance, but the joint energy score, which combines the marginal energy and the latent-guided term, achieves the best performance.
> | AUROC with CIFAR-10 as  in-distribution                          | SVHN | Constant | CIFAR-100 | CelebA |
> |-------------------|------|----------|----------------|--------|
> | $\gamma$=0 | 0.96 | 0.67     | 0.66      | 0.68   |
> | $\gamma$ = 0.01  | 0.98 | 0.87  | 0.78      | 0.74   |
> | $\gamma$ = 0.1| 0.95 | 0.97   | 0.82      | 0.77   |
> | $\gamma$ = 1  | 0.94 | 0.91    | 0.82     | 0.76   |
> | $\gamma$ = 10 | 0.94 | 0.88     | 0.81      | 0.76   |
> | $\gamma$ = 1 with only latent-guided term | 0.94 | 0.87     | 0.81      | 0.76   |
>
> | AUROC with ImageNet 32 as  in-distribution                          | SVHN | Constant | FMNIST | CelebA |
> |-------------------|------|----------|----------------|--------|
> | $\gamma$=0 | 0.99 | 0.97     | 0.40      | 0.48   |
> | $\gamma$ = 0.01  | 0.99 | 0.97  | 0.43      | 0.49   |
> | $\gamma$ = 0.1| 0.99 | 0.99   | 0.64      | 0.51   |
> | $\gamma$ = 1  | 0.98 | 0.99    | 0.90     | 0.52   |
> | $\gamma$ = 10 | 0.91 | 0.95     | 0.91      | 0.52   |
> | $\gamma$ = 1 with only latent-guided term | 0.87 | 0.82     | 0.91      | 0.52   |
> Note: only latent-guided term means $s(\mathrm{x}):=\gamma\operatorname{sim}\left(g\_\psi\left(f_\phi(\mathrm{x})\vphantom{f\_\phi}\right), h(\mathrm{x})\right)$
>
> We don't think leveraging SSRL's advantage on semantic representation to improve OOD performance is a weakness.  As shown in Tab.4, relying solely on the marginal energy is far from sufficient for strong OOD detection. On the contrary, we find a robust way to improve OOD ability through self-supervised representations as latent guidance.
> ## Analysis of ERML setting
> We do not claim that our ERML setting suffers from mode collapse or blurry outputs. On the contrary, it may help mitigate potential mode collapse by matching both the energy distribution and the real data distribution, as also discussed in [1,2]. Moreover, by matching the energy distribution, EBM training can compensate for the blurry outputs induced by the ELBO loss[3].
>
> [1] Cui, Jiali, and Tian Han. "Learning energy-based model via dual-MCMC teaching."  NeurIPS 2023.
>
> [2] Geng, Cong, et al. "Improving adversarial energy-based model via diffusion process." ICML 2024.
>
> [3] Luo, Yihong, et al. "Energy-calibrated vae with test time free lunch." ECCV 2024.

---

> > ### Author Rebuttal · Reviewer_ViZV · 2026-04-03
> >
> > Thanks for the rebuttal. I will keep my score

---

> > > ### Author Response · Authors · 2026-04-07
> > >
> > > Thanks for your acknowledgement! Your comments are instrumental in shaping the quality of our paper.
> > >
> > > Our original manuscript and rebuttal response demonstrate the superiority of our method from multiple perspectives, such as generation quality and OOD detection performance, greatly narrowing the gap between EBMs and mainstream generative models. Our latent-guided mechanism helps the energy function capture both the semantic geometry of the data manifold and the data likelihood. Modeling data distributions together with their geometric structure is the foundational motivation and the ultimate objective of generative models, while most existing methods mainly focus on the former. Thus, as a parting remark, we would be grateful if you would consider raising your score in light of our method's potential to advance EBM research and the broader generative modeling.

---

### Official Review · Reviewer_skNS · 2026-03-11

**Soundness:** 2
**Presentation:** 2
**Significance:** 3
**Originality:** 3
**Overall Recommendation:** 4
**Confidence:** 4

**Summary:**

This paper proposes a latent-guided cooperative energy-based model that learns a joint energy over data and semantic latent targets, where the latent targets are constructed from a pretrained self-supervised encoder. The main idea is to use the latent variable to inject semantic structure into EBM learning, while keeping negative-phase sampling in data space only, and to couple the EBM with a generator that is trained to match both the energy model and the real data-latent distribution.

**Compliance With Llm Reviewing Policy:**

Affirmed.

**Key Questions For Authors:**

In the paper, the author defines $q_{data}(z|x)$ using a pretrained self-supervised encoder, but it is unclear why this should be viewed as a true target posterior. It would be good to clarify the modeling assumptions behind this choice and what properties of the pretrained encoder make it suitable for defining the target latent distribution.

One thing that I am confused is that Algorithm 1 appears to use the conjugate exponential-family construction to compute the joint and marginal energies, while the posterior is parameterized separately using cosine similarity (Eqn12). If so, these choices do not seem to come from one consistent probabilistic model, since under the strict conjugate form, the posterior $p(z|x)$ is already determined. It would be good to clarify whether the final implementation mixes these two design choices, or whether they are intended as separate alternatives.

The paper discusses different posterior parameterizations in Eq.11 and Eq.12, but their role in the final training algorithm is not clear. It would be good to clarify how Eq.11 and Eq.12 are used in practice or in leading to later modelling designs, and whether they are part of the final implemented model or only alternative formulations discussed in the text.

**Limitations:**

Please see the Key Questions.

**Strengths And Weaknesses:**

A main strength of the paper is that it has a clear core idea. Using a semantic latent target to guide EBM learning is intuitive, and the key theoretical result in Proposition 3.1 is useful. The paper also includes a good range of experiments, including generation quality, OOD detection, and conditional sampling. The ablation studies are also helpful in supporting the proposed method.

But, the main presentation, especially the theoretical part, is not always clear. Several important modeling choices and derivations are difficult to follow, and some parts of the final method are not presented in a fully consistent way. Please see the Key Questions.

---

> ### Author Rebuttal · Authors · 2026-03-31
>
> Thanks for your valuable review. We address your concerns below.
> ## Choice of Pretrained Self-Supervised Encoder as Target Posterior
> Thanks for your good question! The true target posterior can be defined in multiple ways, since we only have access to observed data. Our core motivation is to preserve semantic features and geometric information in latent space and leverage this latent variable to guide energy training.
>
> Self-supervised representation learning is known to extract semantic features and has recently been shown to improve generative modeling[1,2,3]. Thus, it's natural to choose self-supervised representation as our target latent variable. Our latent-guided EBM enables the energy function to capture both the semantic geometry of the data manifold and the data likelihood, see Fig.1. Modeling data distributions while understanding their geometric structure is both the foundational motivation and the ultimate objective of generative models.
> ## Definition of Joint Energy Forms
> The conjugate exponential family form and cosine-similarity posterior are two separate alternatives. We construct our joint energy distribution from two perspectives. First, the conjugate exponential-family form follows a unified formulation, under which $p_{\phi,\psi}(\mathrm{z|x})$ is already determined. Second, the cosine-similarity posterior is introduced through a separate formulation $ E_\theta(\mathrm{x,z}) = F\left(f_\phi(\mathrm{x})\right)-\log p_{\phi,\psi}(\mathrm{z|x})$, under which $p_{\phi,\psi}(\mathrm{z|x})$ takes a cosine-similarity form. Section 4.5 and Fig.6 provide the analysis and comparisons between these forms.
> ## Clarification of Eqs.11 and 12
> As explained above, if we choose the separate form to define joint energy distribution, we need to define an explicit latent posterior. We consider two choices of this posterior $p_{\phi,\psi}(\mathrm{z|x})$. One is the Gaussian posterior(Eq.11), the other is the cosine-similarity posterior(Eq.12). We choose the cosine-similarity form for our image generation experiments as it gets better performance, as shown in Fig.6.
>
> [1] Yu, Sihyun, et al. "Representation alignment for generation: Training diffusion transformers is easier than you think." ICLR 2025.
>
> [2] Wu, Ge, et al. "Representation Entanglement for Generation: Training Diffusion Transformers Is Much Easier Than You Think." NeurIPS 2025.
>
> [3] Zheng, Boyang, et al. "Diffusion transformers with representation autoencoders." ICLR 2026.

---

> > ### Author Rebuttal · Reviewer_skNS · 2026-04-07
> >
> > My main concern is the clarity of the method, especially in the theoretical presentation. The paper introduces both a conjugate exponential-family formulation and a later separate formulation with an explicit posterior, but it does not clearly explain in the main text which one corresponds to the final implemented model. Although I understand that these are two separately defined alternatives, I find it unclear, even after reading the rebuttal, which formulation is actually used in the final implementation and main experiments. This should be stated much more explicitly in the manuscript or rebuttal.
> >
> > I am also unconvinced by the use of a pretrained self-supervised encoder as the target posterior. While this is intuitively reasonable, it is not clear why it should be viewed as the appropriate target posterior.

---

> > > ### Author Response · Authors · 2026-04-08
> > >
> > > Thanks for your acknowledgement! We apologize for any confusion caused by our presentation in manuscript and rebuttal response. Exactly as you understood it, we define our joint energy function $E_{\theta}(\mathrm{x}, \mathrm{z})$ in both a conjugate exponential-family form and a separate form with an explicit posterior. We use the separate form with cosine-similarity posterior in our final implementation and main experiments, that is, we use Eq.10
> > > $$ E_\theta(\mathrm{x,z}) = F\left(f_\phi(\mathrm{x})\right)-\log p_{\phi,\psi}(\mathrm{z|x}) $$ to define our joint energy function where $$ p_{\phi,\psi}(\mathrm{z|x})=\frac{\exp\left(\gamma \operatorname{sim}\left(g_\psi\left(f_\phi(\mathrm{x})\vphantom{f_\phi}\right), \mathrm{z}\right)\right)}{Z_\gamma}, \quad \mathrm{z}\sim\mathbb{S}^{d_{\mathrm{z}}-1} $$ as in Eq.12.
> > >  We will make all of these clear and more explicit in our updated version. Thank you for helping us strengthen the paper.
> > >
> > > For your second question, we are not entirely sure which specific point is confusing, so we will explain it based on our understanding. First, we should clarify that the 'target' posterior is not an absolute or uniquely defined notion; we relax the prior, so for our 'target' joint distribution, we can theoretically choose $p_{\text{data}}(\mathrm{z|x})$ very flexibly. What we should consider is which choice can leverage useful information to facilitate EBM training and be easy to learn. A pretrained self-supervised encoder is known to extract meaningful semantic features from the data manifold[1,2,3,4,5], which we anticipated could serve as effective latent guidance for energy training. After experimental validation, we found that it does improve generation quality and OOD performance, which confirms the feasibility of our choice. If your question is why a pretrained self-supervised encoder can extract meaningful semantic features, we refer to the references listed above, as this has been widely studied and validated across various tasks. If your question is why latent semantic understanding can improve distribution modeling, this is a long-standing fundamental problem in the generative modeling community and likely requires continued exploration. At a high level, for complex datasets, latent semantic understanding can help a generative model better and faster capture the geometry of the data manifold, on top of which the data distribution can then be modeled more effectively and efficiently.
> > >
> > > Stepping back, we can also understand it from a practical perspective. Our goal is to use the latent-guided mechanism to improve our joint energy on generation and other applications, rather than fit the target posterior perfectly.
> > > That means, even if the pretrained self-supervised encoder is not perfect, the latent-guided information from the target posterior can still leveraged by the marginal backbone $F\left(f_\phi(\mathrm{x})\right)$ to improve generation, and the learned posterior $p_{\phi,\psi}(\mathrm{z|x})$ can help improve OOD detection and other applications. Therefore, our joint energy is trained to capture sufficient statistics and geometric information of the data manifold, rather than to fit a perfect target posterior.
> > >
> > > Of course, we can choose other methods as the target posterior, and we can even attempt to self-learn an appropriate target posterior to obtain meaningful representations that better match the statistical characteristics of the data distribution, but this would also be much more challenging. Overall, our method is just the beginning of latent-guided EBMs; the choices of target posterior and joint energy function can be explored more in the future. We will clarify these in our updated version.
> > >
> > > We hope our response has addressed your concerns. If any point remains unclear or if you have further questions, since there is not much time left for rebuttal, we would kindly suggest that you add them directly in the Acknowledgement window as soon as possible, and we can update our reply accordingly.
> > >
> > > [1] Chen, Ting, et al. "A simple framework for contrastive learning of visual representations." ICML, 2020.
> > >
> > > [2] Grill, Jean-Bastien, et al. "Bootstrap your own latent: A new approach to self-supervised learning." NeurIPS 2020.
> > >
> > > [3] Caron, Mathilde, et al. "Emerging properties in self-supervised vision transformers." ICCV, 2021.
> > >
> > > [4] Yu, Sihyun, et al. "Representation alignment for generation: Training diffusion transformers is easier than you think." ICLR 2025.
> > >
> > > [5]Zheng, Boyang, et al. "Diffusion transformers with representation autoencoders." ICLR 2026.

---

### Official Review · Reviewer_jxKY · 2026-03-12

**Soundness:** 3
**Presentation:** 2
**Significance:** 3
**Originality:** 3
**Overall Recommendation:** 4
**Confidence:** 4

**Summary:**

This paper introduces a novel way to train latent variable energy-based models. The main contributions are claimed improved ways of connecting data and latent space domains during the training of the model via data augmentation, pre-trained encoder networks, generator networks, and MCMC. To my understanding, the following steps are the core of the proposed methodology:

1. As a first step, the authors generate auxiliary latent data via data augmentation. For data point $x$, data augmentation via rotations, cropping, noise, and blurring processes produces the corrupted state $v$. The augmented data point, denoted as $v$, is then mapped to a representation $z$ using a pre-trained SimCLR encoder. The latent data generating process is denoted as $p_{\mathrm{data}}(z \vert x)$.
2. The model for $p_{\mathrm{data}}(x, z)$ is the latent variable energy-based model $\exp(-E(x, z))$. The energy is parametrised, to my understanding, as $E(x, z) = F(f_\phi(x)) - \log p_{\phi, \psi}(z \vert x)$, where $f_\phi(x)$ is a shared neural network in both terms, but the marginal reduces to $E(x) = F(f_\phi(x))$, avoiding numerical marginalisation. The encoder network is parametrised as $\exp(\gamma \mathrm{sim}(g_\psi(f_\phi(x)), z))$.
3. The training of the energy-based model requires self-sampling from $p_\theta(x, z)$. The authors reduce the training loss to the sampling from $p_\theta(x)$. The samples from $p_\theta(x)$ are then obtained from Langevin Monte Carlo iterations as in contrastive divergence. For efficiency reasons, a generator network $G$ is regressed to the outputs of Langevin MCMC, thereby amortising the cost of long-run Markov chains into an additional generator network. This step is called energy matching. To avoid mode collapse, the negative samples from $p_\theta$ undergo random data augmentation as in step 1.
4. Additionally, the authors introduce the distribution $p_g(x, z \vert m) \propto \exp(-||x - G(m)||^2) \exp(\rho sim(z, L(G(m)))$, where $L$ is a learned mapping from the data domain to the latent space domain, similar to those in step 2. $p_g(x, z \vert m) $ is fitted to $p_{data}(x, z)$ using the evidence lower bound, introducing an additional inference network $q_\alpha(m \vert x)$, with $m$ being the Gaussian latent variable of the generator $G$.

Different choices in the described methodology lead to methods called LGCEBM-EM, LGCEBM-EJM, LGCEBM-ERML and LGCEBM-ERME, which are tested extensively on image generation, out of distribution detection, conditional image generation, and image restoration.

**Compliance With Llm Reviewing Policy:**

Affirmed.

**Final Justification:**

This paper provides an interesting exploration of the design space for cooperative training of energy-based models, and my concerns were largely addressed by the rebuttal. However, I believe that the clarity of the paper should be improved for a potential camera ready version. Considering the typical paper quality at ICML, I would like to maintain my score of four, which I believe reflects my assessment of this paper. In a borderline case, I would recommend this paper for acceptance to the AC based on the constructive discussion with the authors.

**Key Questions For Authors:**

- What does EJM, ERME stand for, and what method does LGCEBM†-EM correspond to?
- Given that EM without data augmentation yields the best FID score on CIFAR-10, what applications or types of dataset necessitate data augmentation?
- Is there one modelling setup that you can recommend for image generation, or does this need to be tuned for each dataset?
- Given multiple competing neural networks, how difficult was it to get everything running stably? For a new unknown dataset, does your training method require extensive tuning? I find that one of the nice properties of diffusion and flow matching models is the relative ease of training, and it would be great if stable training methods also existed for EBMs.

**Limitations:**

The limitations were not discussed. The current approach requires many networks playing different roles, and it is not clear from the paper if this is a limiting factor when scaling the approach to larger and larger datasets due to training instability or training cost.

**Strengths And Weaknesses:**

Strengths:
- Extensive experiments, including superior FID scores on large scale benchmark datasets like imagenet.
- Demonstration of performance on various downstream tasks.
- Ablations with various modelling choices and report of used hyper parameters.
- Approach promotes disentanglement on latent space.
- Avoids memory intensive techniques like replay buffers.
- Requires no marginalisation of latent $z$ to obtain energy on data space.
- If I am understanding correctly, the approach enables fast image generation with one generator evaluation and few MCMC refinement steps.

Weaknesses:
- The paper introduces a lot of networks, modelling choices, and notation: Two distinct types of data augmentation (denoted as $v$), two types of latent variable (a semantic one denoted as $z$, and one for the generator, denoted as $m$), three types of encoder (a pretrained SimCLR embedding $h$, an encoder $g_\psi$, trying to imitate structural information of $h$, a variational distribution $q_\alpha$ for the one of the two loss contributions for generator training), three types of self-sampling mechanisms to approximate the negative phase term of the EBM loss (generator $G$, Langevin MCMC steps, random data augmentations). There are furthermore two choices of posterior modelling $p_{data}(z \vert x)$ and a separate modelling choice for $p_g(x, z \vert m)$. The amount of options clutters the presentation. It becomes harder to understand which steps follow from derivations of maximum-likelihood learning of $p_{ebm}$, and which steps are (at least partially) heuristics to boost performance. The derived methods have many different names: LGCEBM-EM w/o MCMC, LGCEBM-EM
LGCEBM-ERML w/o MCMC, LGCEBM-ERML, LGCEBM†-EM, LGCEBM-EJM, LGCEBM-ERME etc.. I couldn't figure out what EJM and ERME stand for, to memorise them more easily while reading. The paper would be vastly improved by picking a central story, and introducing alternative approaches and their ablations in later subsections or in the appendix. Finally, the EM setting without data augmentation appears to achieve the best results on CIFAR-10 in FID according to figure 7, which suggests that a simplified methodology would suffice.
- Adding to the previous point, several of the introduced methods perform very close to each other. The hyperparameter and architecture choices may therefore be more consequential than the modelling choices.

---

> ### Author Rebuttal · Authors · 2026-03-31
>
> Thanks for your valuable review. We address your concerns below.
> ## Notation Clarification
> EJM and ERME denote different generator training schemes.
>
> EJM denotes training our generator to match the energy distribution with MCMC refinement under the joint energy function, i.e., using Eqs.14 and 16.
>
> ERME denotes training our generator to match both the energy
> and real data distribution&thinsp;(i.e. ERM setting using Eq.17), and choosing energy form $g\_{\psi}\left(f_{\phi}(G_\zeta(\mathrm{m}))\vphantom{f_\phi}\right)$ as $L(G_\zeta(\mathrm{m}))$ when defining $p\_g(\mathrm{x,z|m})$ in Eq.19.
> We denote these names in Section 4&thinsp;(lines 249-251).
>
> LGCEBM$^\dagger$-EM adopts the basic EM setting, where the generator is trained to match the energy distribution with MCMC under the marginal energy, i.e., using Eqs.14 and 15, but with modified architectures and refined optimizer settings, as described in Section 4.1 and Appendix F. We will further clarify these names and notations in our final paper.
>
> To facilitate a better understanding of our method, we provided an algorithm description in Appendix E.
> ## Necessity of Data Augmentation
> Our data augmentation is primarily used to mitigate the energy's catastrophic forgetting in distant regions that always exist in persistent and short-run MCMC EBMs[1,2]. As in Tab.6, our data augmentation enhances OOD detection for distant outliers like the Constant Dataset. We design this for a better energy landscape with minimal effect on sampling quality. Notably, we also tried replay buffers[3] and real samples initialization[4], which are commonly used in short-run EBMs, but only our augmentation strategy proved effective for cooperative training and is simpler for implementation. We further verify the benefit of stochastic augmentation for OOD detection on ImageNet 32 below.
> Method                       | SVHN | Constant | FMNIST | CelebA |
> |-------------------|------|----------|----------------|--------|
> | w/o Aug | 0.89 | 0.84    | 0.85     | 0.41   |
> | w/ Aug | 0.98 | 0.99  | 0.90      | 0.52   |
>
> ## Modeling Setup Recommended for Image Generation
> Thanks for your constructive comments! For image generation, we recommend the EM setting, as it is the simplest and performs well across all evaluated datasets. The ERML setting is used for reconstruction-based applications. In this case, the autoencoder structure enables efficient acquisition of strong reconstruction initializations.
> ## Stability of Our Method
> Thanks for your good question! Training instability has long been a major challenge for EBMs. But we didn't observe empirical signs of instability in our method.
> Fig.7 demonstrates that our latent-guided joint space optimization significantly stabilizes training. The modified version has also been tested with multiple energy-function networks and remains consistently stable. Please see the results below.
> | Method             | FID↓ | Params(M)↓ | NFE↓ |
> |--------------------|------|------------|------|
> | EC-VAE             | 5.20 | 101        | 1    |
> | CDRL               | 4.31 | 73         | 96   |
> | CDRL-Large         | 3.68 | 177        | 96   |
> | Ours(EC-VAE)       | 4.28 | 46         | 11   |
> | Ours(CDRL)         | 3.87 | 56         | 16   |
> | Ours        | 3.28 | 34         | 16   |
> Note: Ours(*) denotes replacing our energy function with that of CDRL or EC-VAE.
> ## Training Cost
> We have shown the training cost below for comparison.
> | Method            | Memory usage(GB)↓ | Training iterations(K)↓ | FID↓ |
> |-------------------|------------------:|------------------------:|-----:|
> | R3GAN          | 57               | 195                   | 3.64 |
> | VAEBM             | 129               | 25                      | 12.2 |
> | CLEL              | 10                | 100                     | 8.61 |
> | CDRL              | 69                | 400                     | 4.31 |
> | LGCEBM-EM (ours)   | 14                | 100                     | 4.23 |
> | LGCEBM$^\dagger$-EM (ours)   | 24                | 100                     | 3.28 |
>
> [1] Abdulsalam, Azwar, and Joseph G. Makin. "Revisiting Contrastive Divergence for Density Estimation and Sample Generation." TMLR 2025.
>
> [2] Nijkamp, Erik, et al. "On the anatomy of mcmc-based maximum likelihood learning of energy-based models." AAAI 2020.
>
> [3] Yilun Du, et al. “Improved Contrastive Divergence Training of Energy-Based Model” ICML 2021.
>
> [4] Geoffrey E. Hinton, “Training Products of Experts by Minimizing Contrastive Divergence.” Neural Computation 2002.

---

> > ### Author Rebuttal · Reviewer_jxKY · 2026-04-03
> >
> > Thank you for your detailed response and the additional statistics. My questions were addressed. I would urge the authors in a revision of their work to introduce all acronyms, ideally with what each letter corresponds to (e.g. EJM (Energy-Joint Matching)). I noticed the descriptions of EJM and ERME in lines 249-251, but as non-experts of your work you could make it easier for the reader to quickly map design-choice - acronym - result. Similarly, while Algorithm 1. may contain everything one needs to know, it is not self-evident what choices in Alg 1. would yield LGCEBM-ERML w/o MCMC, for example. Even in this rebuttal, NFE is not introduced, and I have to guess its meaning.

---

> > > ### Author Response · Authors · 2026-04-03
> > >
> > > Thanks for your acknowledgement! We completely agree that introducing all notations and acronyms more clearly will make the paper easier for readers to follow. We will make sure to revise the manuscript accordingly. We also apologize for any confusion caused by our use of 'NFE' notation. Thank you for helping us strengthen the paper.

---

### Decision · Program_Chairs · 2026-04-30

**Decision:**

Accept (regular)

**Comment:**

This paper proposes Latent-Guided Cooperative Energy-Based Models (LGCEBM), a framework that incorporates semantic latent variables derived from a pre-trained encoder to guide the training of energy-based models. By formulating a joint energy over data and latent space while restricting MCMC sampling to the data domain, the method improves training efficiency and stability. The approach is further coupled with a generator network to amortize sampling and enhance generation quality. Empirical results are shown to justify the effectiveness of the method.

All reviewers agree that the paper presents a meaningful and well-motivated contribution to the development of EBMs. The rebuttal has addressed major concerns regarding lack of clarity, lack of limitation analysis, missing related work, notation inconsistencies, missing baseline comparisons and training cost analysis. After the rebuttal and discussion, the reviewers reached a consensus in favor of a weak accept. The AC considers this work a valuable contribution to the field of EBMs, with both theoretical and empirical significance. The authors are encouraged to further strengthen the paper by incorporating additional results and discussions from the rebuttal into the final version.